# Last-Iterate Convergence of General Parameterized Policies in Constrained MDPs

**Washim Uddin Mondal**                                                  *wmondal@iitk.ac.in*
*IIT Kanpur*

**Vaneet Aggarwal**                                                      *vaneet@purdue.edu*
*Purdue University*

**Reviewed on OpenReview:** *https://openreview.net/forum?id=JedrMCZC6l*

## Abstract

This paper focuses on learning a Constrained Markov Decision Process (CMDP) via general parameterized policies. We propose a Primal-Dual based Regularized Accelerated Natural Policy Gradient (PDR-ANPG) algorithm that uses entropy and quadratic regularizers to reach this goal. For parameterized policy classes with a transferred compatibility approximation error, $\epsilon_{\text{bias}}$, PDR-ANPG achieves a last-iterate $\epsilon$ optimality gap and $\epsilon$ constraint violation with a sample complexity of $\tilde{\mathcal{O}}(\epsilon^{-2}\min\{\epsilon^{-2}, \epsilon_{\text{bias}}^{-\frac{1}{3}}\})$. If the class is incomplete ($\epsilon_{\text{bias}} > 0$), then the sample complexity reduces to $\tilde{\mathcal{O}}(\epsilon^{-2})$ for $\epsilon < (\epsilon_{\text{bias}})^{\frac{1}{6}}$. Moreover, for complete policies with $\epsilon_{\text{bias}} = 0$, our algorithm achieves a last-iterate $\epsilon$ optimality gap and $\epsilon$ constraint violation with $\tilde{\mathcal{O}}(\epsilon^{-4})$ sample complexity. It is a significant improvement over the state-of-the-art last-iterate guarantees of general parameterized CMDPs.

## 1 Introduction

Constrained Markov Decision Process (CMDP) is a classical framework where an agent repeatedly interacts with an unknown environment to maximize the cumulative discounted rewards while simultaneously ensuring that the cumulative observed costs are within a pre-defined boundary. It finds its application in a multitude of practical scenarios. For example, consider an autonomous vehicle that attempts to reach its destination via the shortest-time route without violating traffic rules or a corporate leader who aims to maximize revenue without crossing a monetary budget. In these cases, any departure from the boundary set by the predefined rules can be signaled by a cost while the progress towards the desired objective can be indicated by a reward.

Finding an optimal policy to navigate an unknown CMDP is a difficult task. Nevertheless, several recent articles have proposed algorithms to solve this challenging problem with optimality guarantees. For example, Mondal & Aggarwal (2024a) exhibited that their primal dual-based algorithm can achieve $\epsilon$ optimality gap and $\epsilon$ constrained violation with a $\tilde{O}(\epsilon^{-2})$ sample complexity. Unfortunately, the majority of these works define constraint violation in an average sense. In other words, if their algorithms yield $\{\pi_1, \cdots, \pi_K\}$ policies in $K$ iterations, then the violation is defined as that experienced by a uniformly chosen policy. Since the very nature of this definition allows a large violation at one iteration to get balanced by a smaller violation later, such algorithms are not suitable for safety-critical applications.

To address this challenge, some recent articles have proposed algorithms with last-iterate guarantees. For example, Gladin et al. (2023); Ying et al. (2022) prove last-iterate guarantees for softmax policies, whereas Ding et al. (2024) establishes the same for log-linear policies. Montenegro et al. (2024) recently achieved $\tilde{\mathcal{O}}(\epsilon^{-7})$ sample complexity via general parameterized policies. It is to be emphasized that general parameterization subsumes the tabular softmax and log-linear cases and allows the policies to be represented by neural networks. Moreover, since the general parameterization uses a fixed d number of parameters where d is independent of the size of the state space, it can also be utilized for large or infinite state space. Notably,

| Algorithm | Sample Complexity | Parameterization |
|-----------|-------------------|------------------|
| Dual Descent (Ying et al., 2022) | $\tilde{\mathcal{O}}(\epsilon^{-2})$ | Softmax |
| Cutting-Plane (Gladin et al., 2023) | $\tilde{\mathcal{O}}(\epsilon^{-4})$ | Softmax |
| RPG-PD (Ding et al., 2024) | $\tilde{\mathcal{O}}(\epsilon^{-6})$ | Log-linear with $\epsilon_{\text{bias}} = \mathcal{O}(\epsilon^8)$ |
| C-PG (Montenegro et al., 2024) | $\tilde{\mathcal{O}}(\epsilon^{-7})$ | General |
| PDR-ANPG (**This Work**) | $\tilde{\mathcal{O}}(\epsilon^{-4})$ | General with $\epsilon_{\text{bias}} = 0$ |
| PDR-ANPG (**This Work**) | $\tilde{\mathcal{O}}(\epsilon^{-2}\min\{\epsilon^{-2}, \epsilon_{\text{bias}}^{-\frac{1}{3}}\})$ | General with $\epsilon_{\text{bias}} > 0$ |
| Lower Bound (Vaswani et al., 2022) | $\Omega(\epsilon^{-2})$ | – |

Table 1: List of recent papers on CMDPs with last-iterate guarantees. The term $\epsilon_{\text{bias}}$ is the expressivity error of the underlying parameterized policy class.

the current-state-of-the-art sample complexity $\tilde{\mathcal{O}}(\epsilon^{-7})$ is far from the lower bound $\Omega(\epsilon^{-2})$. This raises the following question.

> Is it possible to design an algorithm for general parameterized CMDPs whose last-iterate guarantee is better than the current state-of-the-art?

## 1.1 Contribution and Challenges

This paper affirmatively answers the above question. In particular, we propose a primal dual-based regularized accelerated natural policy gradient (PDR-ANPG) algorithm that uses entropy and quadratic regularizer in the Lagrangian function and momentum-based accelerated stochastic gradient descent (ASGD) process of (Jain et al., 2018) as the NPG finding subroutine. We establish that, for general parameterized policies, our algorithm achieves $\mathcal{O}(\epsilon + (\epsilon_{\text{bias}})^{\frac{1}{6}})$ last-iterate optimality gap and the same constraint violation with a sample complexity of $\tilde{\mathcal{O}}(\epsilon^{-2}\min\{\epsilon^{-2}, \epsilon_{\text{bias}}^{-\frac{1}{3}}\})$ where $\epsilon_{\text{bias}}$ indicates the transferred compatibility approximation error of the policy class. If $\epsilon_{\text{bias}} > 0$ is independent of $\epsilon$, the sample complexity is $\tilde{\mathcal{O}}(\epsilon^{-2})$ for small $\epsilon$. However, if $\epsilon_{\text{bias}} = 0$, i.e., the policy class is complete, then the optimality gap and constraint violation are $\mathcal{O}(\epsilon)$ and the sample complexity turns out to be $\tilde{\mathcal{O}}(\epsilon^{-4})$.

One of the main challenges in handling an entropy regularizer is that the advantage estimate can, in general, become unbounded. This is a problem since many intermediate lemmas crucial in establishing global convergence utilize its boundedness. In the tabular setup, Ding et al. (2024) circumvented this problem by allowing the policy optimization to run over a carefully chosen simplex. Such provisions are not available for general parameterized policies. Interestingly, we observe that the advantage estimates can be bounded in an average sense, which is sufficient to obtain our desired result, provided that the gradient[1] sampling is done following an unconventional method. In particular, on top of the standard routines, our sampling process (Algorithm 1) comprise an additional expectation that reduces the variance of the gradient estimate while preserving its unbiasedness.

Our improved sample complexity originates from another important observation. We noticed that the bias of the NPG estimator can be interpreted as the convergence error of an ASGD program with an exact gradient oracle. Without this reduction, the bias would be exponentially large, leading to a significant deterioration in sample complexity.

## 1.2 Related Works

**Unconstrained MDP:** Many algorithms are available in the literature that solve the unconstrained MDP via an exact gradient oracle e.g., see (Zhan et al., 2023; Lan, 2023; Cen et al., 2022; Agarwal et al., 2021; Bhandari & Russo, 2021). Among the sample-based methods, some papers demonstrate first-order convergence (Shen et al., 2019; Huang et al., 2020; Gargiani et al., 2022; Xu et al., 2020) while other works focus on

---

[1]Here, we specifically refer to the gradients used in the NPG finding subroutine.

global convergence (Masiha et al., 2022; Liu et al., 2020; Khodadadian et al., 2022; Chen & Maguluri, 2022). It is to be observed that Fatkhullin et al. (2023); Mondal & Aggarwal (2024b) prove the state-of-the-art $\tilde{\mathcal{O}}(\epsilon^{-2})$ sample complexity for last-iterate and average global convergence respectively. However, to the best of our understanding, the approach of Fatkhullin et al. (2023) is not extendable to CMDPs.

**Constrained MDP:** Most of the CMDP works show average or regret-type constraint violation guarantees. Many among them design tabular model-based (He et al., 2021; Ding et al., 2021; Liu et al., 2021a; Efroni et al., 2020) and model-free (Wei et al., 2022; Bai et al., 2022; Ding et al., 2021) algorithms. Some literature focus on parameterized policies. For example, Liu et al. (2021b); Zeng et al. (2022) deal with softmax policies while Xu et al. (2021); Bai et al. (2023); Ding et al. (2020) handle the general parameterization. The optimal $\tilde{\mathcal{O}}(\epsilon^{-2})$ sample complexity for general parameterized CMDPs is proven by Mondal & Aggarwal (2024a). In comparison, the literature on last-iterate guarantees is relatively nascent. As shown in Table 1, Ying et al. (2022); Gladin et al. (2023) respectively establish $\tilde{\mathcal{O}}(\epsilon^{-2})$ and $\tilde{\mathcal{O}}(\epsilon^{-4})$ sample complexity for softmax policies whereas Ding et al. (2024) achieve $\tilde{\mathcal{O}}(\epsilon^{-6})$ sample complexity for log-linear policies assuming its expressivity error to be $\epsilon_{\text{bias}} = \mathcal{O}(\epsilon^8)$. Recently, Montenegro et al. (2024) exhibited $\tilde{\mathcal{O}}(\epsilon^{-7})$ sample complexity for general parameterization. We show an improvement over this state-of-the-art.

## 2 Notations

In the following table, we summarize the major notations used in the paper for easy reference.

| Notation | Definition |
|---|---|
| $Q_g^\pi,\ V_g^\pi,\ J_g^\pi$ | Value functions corresponding to policy $\pi$ and utility $g$ |
| $A_g^\pi$ | Advantage function corresponding to policy $\pi$ and utility $g$ |
| $\theta,\ \lambda$ | Primal and dual parameters |
| $d^\pi,\ \nu^\pi$ | Occupancy measures corresponding to policy $\pi$ |
| $\mathcal{L}_\tau$ | Lagrangian with regularization parameter $\tau$ |
| $F(\theta)$ | Fisher matrix defined in (12) |
| $\mathcal{E}_\nu^\tau$ | Error function defined in (17) |
| $L_{\tau,\lambda}^2\ \sigma_{\tau,\lambda}^2$ | Constants defined in (16) and (34) respectively |
| $\hat{\hat{\zeta}}_{\theta,\lambda}^\tau$ | Error function defined in (35) |
| $\epsilon_{\text{bias}}$ | Transferred compatibility approximation error |

Table 2: Major notations used in this paper.

## 3 Formulation

Consider a Constrained Markov Decision Process (CMDP) characterized as $\mathcal{M} = (\mathcal{S}, \mathcal{A}, r, c, P, \gamma, \rho)$ where $\mathcal{S}$ is a (possibly infinite) state space, $\mathcal{A}$ is a finite action space with cardinality $A$, $r : \mathcal{S} \times \mathcal{A} \to [0, 1]$ indicates the reward function, $c : \mathcal{S} \times \mathcal{A} \to [-1, 1]$ is the cost function, $P : \mathcal{S} \times \mathcal{A} \to \Delta(\mathcal{S})$ is the transition function (where $\Delta(\cdot)$ denotes the probability simplex on its argument set), $\gamma \in [0, 1)$ defines the discount factor and $\rho \in \Delta(\mathcal{S})$ is the initial state distribution. This paper assumes $\mathcal{S}$ to be countable, though, in general, it can be taken to be compact. A (stationary) policy is defined to be a function of the form $\pi : \mathcal{S} \to \Delta(\mathcal{A})$. For a given policy, $\pi$, and a state-action pair $(s, a)$, the $Q$ value corresponding to a utility function $g : \mathcal{S} \times \mathcal{A} \to \mathbb{R}$ is defined as follows (the term utility function subsumes the concepts of both reward and cost functions).

$$Q_g^\pi(s, a) = \mathbf{E}_\pi \left[ \sum_{t=0}^{\infty} \gamma^t g(s_t, a_t) \middle| s_0 = s, a_0 = a \right] \tag{1}$$

where $\mathbf{E}_\pi$ is the expectation over all $\pi$-induced trajectories $\{(s_t, a_t)\}_{t=0}^{\infty}$ where $a_t \sim \pi(s_t)$, $s_{t+1} \sim P(s_t, a_t)$, $\forall t \in \{0, 1, \cdots\}$. Similarly, given a policy $\pi$ and utility function $g$, the associated state value function is given as: $V_g^\pi(s) = \sum_a \pi(a|s) Q^\pi(s, a)$, $\forall s \in \mathcal{S}$.

Moreover, for a given policy $\pi$, and a utility function $g$, the advantage function is: $A_g^\pi(s,a) = Q_g^\pi(s,a) - V_g^\pi(s)$, $\forall(s,a)$. The state occupancy measure corresponding to $\pi$ is given as

$$d^\pi(s) = (1-\gamma)\mathrm{E}_\pi\left[\sum_{t=0}^\infty \gamma^t \mathbf{1}(s_t = s)\Big| s_0 \sim \rho\right], \ \forall s \tag{2}$$

where $\mathbf{1}(\cdot)$ is the indicator function. We ignore the dependence on $\rho$ for simplifying the notations whenever there is no confusion. The state-action occupancy measure induced by $\pi$ is given as: $\nu^\pi(s,a) = d^\pi(s)\pi(a|s)$, $\forall(s,a)$. We define $\mathbf{E}_{s\sim\rho}[V_g^\pi(s)] = J_g^\pi$. The goal of learning CMDP is to solve the following optimization.

$$\max_{\pi\in\Pi} J_r^\pi \ \text{ subject to: } J_c^\pi \geq 0 \tag{3}$$

where $\Pi$ is the set of all policies. We assume that at least one interior solution exists for the above optimization, which is known as Slater's condition.

**Assumption 1.** *There exists $\bar{\pi} \in \Pi$ such that $J_c(\bar{\pi}) \geq c_{\text{slat}}$ where $c_{\text{slat}} \in (0, 1/(1-\gamma)]$.*

Note that the policy function cannot be expressed in a tabular format for infinite states. General parameterization can be used in such cases. It indexes each policy by a d-dimensional parameter $\theta$. Define $J_g^{\pi_\theta} \triangleq J_g(\theta)$ for any $g \in \mathbb{R}^{\mathcal{S}\times\mathcal{A}}$. Problem (3) can now be written as:

$$\max_{\theta\in\mathbb{R}^d} J_r(\theta) \ \text{ subject to: } J_c(\theta) \geq 0 \tag{4}$$

## 4    Algorithm Design

The standard approach to solve the constrained optimization equation 3 is utilizing a saddle point optimization on the Lagrangian function $J_{r+\lambda c}^\pi$ where $\lambda$ is a Lagrange multiplier. We utilize $\pi^*$ to denote an optimal solution to equation 3. Moreover, $\lambda^*$ is its corresponding dual solution.

$$\lambda^* \in \arg\min_{\lambda\geq 0}\max_{\pi\in\Pi} J_{r+\lambda c}^\pi \tag{5}$$

The following result is well known (Ding et al., 2024).

**Lemma 1.** *An optimal primal-dual pair $(\pi^*, \lambda^*)$ is guaranteed to exist if Assumption 1 holds. Moreover, it satisfies the following strong duality condition.*

$$\max_{\pi\in\Pi} J_{r+\lambda^* c}^\pi = J_{r+\lambda^* c}^{\pi^*} = \min_{\lambda\geq 0} J_{r+\lambda c}^{\pi^*} \tag{6}$$

*Additionally, $0 \leq \lambda^* \leq 1/[(1-\gamma)c_{\text{slat}}]$.*

This paper, however, considers a regularized Lagrangian function, defined below $\forall\pi \in \Pi$, $\forall\lambda \geq 0$.

$$\mathcal{L}_\tau(\pi,\lambda) = J_{r+\lambda c}^\pi + \tau\left(\mathcal{H}(\pi) + \frac{1}{2}\lambda^2\right) \tag{7}$$

where $\tau$ is a tunable parameter and $\mathcal{H}(\pi)$ is the entropy corresponding to the policy $\pi$ (defined below).

$$\mathcal{H}(\pi) \triangleq -\frac{1}{1-\gamma}\sum_{s,a} d^\pi(s)\pi(a|s)\log\pi(a|s) = \mathbf{E}_\pi\left[\sum_{t=0}^\infty -\gamma^t\log\pi(a_t|s_t)\Big| s_0 \sim \rho\right] \tag{8}$$

Let $(\pi_\tau^*, \lambda_\tau^*)$ denote the primal-dual solutions corresponding to the regularized Lagrangian $\mathcal{L}_\tau$, i.e.,

$$\begin{aligned}\pi_\tau^* &\triangleq \arg\max_{\pi\in\Pi}\min_{\lambda\in\Lambda}\mathcal{L}_\tau(\pi,\lambda),\\ \lambda_\tau^* &\triangleq \arg\min_{\lambda\in\Lambda}\max_{\pi\in\Pi}\mathcal{L}_\tau(\pi,\lambda)\end{aligned} \tag{9}$$

where $\Lambda = [0, \lambda_{\max}]$ is a carefully chosen set of positive reals and $\lambda_{\max}$ is stated in Theorem 1. We can prove that for any $\tau, \lambda_{\max} > 0$, the pair $(\pi^*_\tau, \lambda^*_\tau)$ uniquely exists and follows a strong duality similar to Lemma $1^2$. However, such a result cannot be directly applied to the class of parameterized policies where our objective is to solve (10) where $\mathcal{L}_\tau(\pi_\theta, \lambda)$ is denoted as $\mathcal{L}_\tau(\theta, \lambda)$.

$$\min_{\lambda \in \Lambda} \max_{\theta \in \mathbb{R}^d} \mathcal{L}_\tau(\theta, \lambda) \tag{10}$$

We aim to apply the Natural Policy Gradient (NPG)-based primal-dual updates (expressed below) to solve (10), starting with arbitrary $\theta_0$ and $\lambda_0$.

$$\begin{aligned} \theta_{k+1} &= \theta_k + \eta F(\theta_k)^\dagger \nabla_\theta \mathcal{L}_\tau(\theta_k, \lambda_k), \\ \lambda_{k+1} &= \mathcal{P}_\Lambda \left[ (1 - \eta\tau)\lambda_k - \eta J_c(\theta_k) \right] \end{aligned} \tag{11}$$

where $\mathcal{P}_\Lambda$ denotes the projection operation onto $\Lambda$ and $\eta$ is the learning rate. Observe that, unlike the vanilla policy gradient iterations, the update direction of $\theta$ does not align with the gradient $\nabla_\theta \mathcal{L}_\tau(\theta, \lambda)$ but rather, it is modulated by the Moore-Penrose pseudoinverse (denoted as $\dagger$) of the Fisher matrix, $F(\theta)$ defined below.

$$F(\theta) = \mathbf{E}_{(s,a) \sim \nu^{\pi_\theta}} \left[ \nabla_\theta \log \pi_\theta(a|s) \otimes \nabla_\theta \log \pi_\theta(a|s) \right] \tag{12}$$

where $\otimes$ denotes the outer product. The lemma stated below outlines a procedure to compute the gradient $\nabla_\theta \mathcal{L}_\tau(\theta, \lambda)$.

---

**Algorithm 1** Sampling Procedure

1: **Input:** $\theta, \omega, \lambda, \gamma, \tau, r, c, \rho$
2: **Define:** $g \triangleq r + \lambda c + \tau \psi_\theta$

3: $T \sim \mathrm{Geo}(1 - \gamma)$, $s_0 \sim \rho$, $a_0 \sim \pi_\theta(s_0)$
4: **for** $j \in \{0, \cdots, T-1\}$ **do**
5: $\quad s_{j+1} \sim P(s_j, a_j)$ and $a_{j+1} \sim \pi_\theta(s_{j+1})$
6: $\hat{J}_c(\theta) \leftarrow \sum_{j=0}^{T} c(s_j, a_j), \quad \hat{s} \leftarrow s_T$

7: $\qquad\qquad\qquad\qquad\qquad\qquad\qquad\qquad\qquad\qquad\qquad\qquad\qquad$ ▷ Value Function Estimation
8: $T \sim \mathrm{Geo}(1 - \gamma)$, $s_0 \leftarrow \hat{s}$, $a_0 \sim \pi_\theta(s_0)$
9: **for** $j \in \{0, \cdots, T-1\}$ **do**
10: $\quad s_{j+1} \sim P(s_j, a_j)$, and $a_{j+1} \sim \pi_\theta(s_{j+1})$
11: $\hat{V}_g^{\pi_\theta}(\hat{s}) \leftarrow \sum_{j=0}^{T} g(s_j, a_j)$

12: $\qquad\qquad\qquad\qquad\qquad\qquad\qquad\qquad\qquad\qquad\qquad\qquad$ ▷ Q and Advantage Estimation
13: **for** $a \in \mathcal{A}$ **do**
14: $\quad T \sim \mathrm{Geo}(1 - \gamma)$, $(s_0, a_0) \leftarrow (\hat{s}, a)$
15: $\quad$ **for** $j \in \{0, \cdots, T-1\}$ **do**
16: $\qquad s_{j+1} \sim P(s_j, a_j)$, and $a_{j+1} \sim \pi_\theta(s_{j+1})$
17: $\quad \hat{Q}_g^{\pi_\theta}(\hat{s}, a) \leftarrow \sum_{j=0}^{T} g(s_j, a_j)$
18: $\quad \hat{A}_g^{\pi_\theta}(\hat{s}, a) \leftarrow \hat{Q}_g^{\pi_\theta}(\hat{s}, a) - \hat{V}_g^{\pi_\theta}(\hat{s})$

19: $\qquad\qquad\qquad\qquad\qquad\qquad\qquad\qquad\qquad\qquad\qquad\qquad$ ▷ Gradient Estimation

$$\begin{aligned} \hat{F}(\theta) &\leftarrow \mathbf{E}_{a \sim \pi_\theta(\hat{s})} \left[ \nabla_\theta \log \pi_\theta(a|\hat{s}) \otimes \nabla_\theta \log \pi_\theta(a|\hat{s}) \right] \\ \hat{H}_\tau(\theta, \lambda) &\leftarrow \mathbf{E}_{a \sim \pi_\theta(\hat{s})} \left[ \hat{A}_g^{\pi_\theta}(\hat{s}, a) \nabla_\theta \log \pi_\theta(a|\hat{s}) \right] \\ \hat{\nabla}_\omega \mathcal{E}_\nu^{\tau \pi_\theta}(\omega, \theta, \lambda) &\leftarrow \hat{F}(\theta)\omega - \frac{1}{1 - \gamma} \hat{H}_\tau(\theta, \lambda) \end{aligned} \tag{13}$$

20: **Output:** $\hat{J}_c(\theta), \hat{\nabla}_\omega \mathcal{E}_\nu^{\tau \pi_\theta}(\omega, \theta, \lambda)$

---

$^2$More details are available in the appendix.

**Lemma 2.** *The following holds* $\forall \theta \in \mathbb{R}^{\mathrm{d}}, \ \forall \lambda \in \Lambda.$

$$\nabla_\theta \mathcal{L}_\tau(\theta, \lambda) = \frac{1}{1-\gamma} H_\tau(\theta, \lambda), \tag{14}$$

$$H_\tau(\theta, \lambda) \triangleq \mathbf{E}_{(s,a)\sim\nu^{\pi_\theta}} \left[ A_{r+\lambda c+\tau\psi_\theta}^{\pi_\theta}(s, a) \nabla_\theta \log \pi_\theta(a|s) \right] \tag{15}$$

*where* $\psi_\theta(s, a) = -\log \pi_\theta(a|s), \ \forall (s, a).$

The above result is similar to the well-known policy gradient theorem (Sutton et al., 1999). It is worthwhile to mention here that, since $\psi_\theta$ is unbounded, the advantage function $A_{r+\lambda c+\tau\psi_\theta}^{\pi_\theta}$ can, in general, be unbounded. This is quite disadvantageous since many intermediate results in the convergence analysis require the advantage function to be bounded. Fortunately, Lemma 3 (stated below) proves that advantage values can be bounded in an average sense, which turns out to be sufficient for our analysis.

**Lemma 3.** *The following holds* $\forall \theta, \lambda, \tau, s.$

$$\mathbf{E}_{a\sim\pi_\theta(s)} \left[ \left| A_{r+\lambda c+\tau\psi_\theta}^{\pi_\theta}(s, a) \right|^2 \right] \leq L_{\tau,\lambda}^2 \ \text{where} \ L_{\tau,\lambda}^2 \triangleq \frac{8(1+\lambda)^2}{(1-\gamma)^2} + \tau^2 \left[ \frac{32}{e^2} A + \frac{12(\log A)^2}{(1-\gamma)^2} \right] \tag{16}$$

Let $\omega_\tau^*(\theta, \lambda)$ denote the natural policy gradient (NPG), i.e., $\omega_\tau^*(\theta, \lambda) \triangleq F(\theta)^\dagger \nabla_\theta \mathcal{L}_\tau(\theta, \lambda)$. Define the following.

$$\mathcal{E}_\nu^\tau(\omega, \theta, \lambda) = \frac{1}{2} \mathbf{E}_{(s,a)\sim\nu} \left[ \left( \omega^{\mathrm{T}} \nabla_\theta \log \pi_\theta(a|s) - \frac{1}{1-\gamma} A_{r+\lambda c+\tau\psi_\theta}^{\pi_\theta}(s, a) \right)^2 \right] \tag{17}$$

where $\nu \in \Delta(\mathcal{S} \times \mathcal{A})$ and $\omega \in \mathbb{R}^{\mathrm{d}}$. Lemma 2 shows that $\omega_\tau^*(\theta, \lambda) = \arg\min_{\omega \in \mathbb{R}^{\mathrm{d}}} \mathcal{E}_{\nu^{\pi_\theta}}^\tau(\omega, \theta, \lambda)$. This allows us to compute the NPG via a gradient-descent-based iterative algorithm. Observe that,

$$\nabla_\omega \mathcal{E}_{\nu^{\pi_\theta}}^\tau(\omega, \theta, \lambda) = F(\theta)\omega - \frac{1}{1-\gamma} H_\tau(\theta, \lambda) \tag{18}$$

Unfortunately, obtaining the exact gradient given by (18) might not be viable due to the lack of knowledge of the transition function $P$ (which implies a lack of knowledge of the advantage function $A_{r+\lambda c+\tau\psi_\theta}^{\pi_\theta}$ and hence, that of $H_\tau(\theta, \lambda)$). It is also clear from (11) that the update of dual variables requires an exact knowledge of $J_c(\theta_k)$, which is impossible to obtain without knowing $P$. Algorithm 1 provides sample-based estimates of the above quantities.

For a given set of parameters $(\theta, \lambda, \tau)$, Algorithm 1 starts with $s_0 \sim \rho$ and rolls out a $\pi_\theta$ induced trajectory of length[3] $T \sim \mathrm{Geo}(1-\gamma)$. The total cost of this trajectory is assigned as $\hat{J}_c(\theta)$. The terminal state $s_T$ can be shown to behave as a sample $\hat{s}$ taken from $d^{\pi_\theta}$. Let $g \triangleq r + \lambda c + \tau\psi_\theta$. Another $\pi_\theta$-induced trajectory of length $T \sim \mathrm{Geo}(1-\gamma)$ is rolled out next, with $\hat{s} = s_T$ as its starting point. The total sum of utility observed in this trajectory is assigned as the value estimate $\hat{V}_g^{\pi_\theta}(\hat{s})$. To calculate the $Q$ estimates, for each $a \in \mathcal{A}$, a $\pi_\theta$ induced trajectory of length $T \sim \mathrm{Geo}(1-\gamma)$ is rolled out with $(\hat{s}, a)$ as the initial state-action. The total utilities observed in these trajectories are used as the estimates $\{\hat{Q}_g^{\pi_\theta}(\hat{s}, a)\}_{a\in\mathcal{A}}$. Its corresponding advantage values are estimated as $\hat{A}_g^{\pi_\theta}(\hat{s}, a) = \hat{Q}_g^{\pi_\theta}(\hat{s}, a) - \hat{V}_g^{\pi_\theta}(\hat{s}), \ \forall a$. Finally, the Fisher matrix is estimated as the expected value of $\nabla_\theta \log \pi_\theta(a|\hat{s}) \otimes \nabla_\theta \log \pi_\theta(a|\hat{s})$ over $a \sim \pi_\theta(\hat{s})$, while the estimate of $H_\tau(\theta, \lambda)$ is computed as the expectation of $\hat{A}_g^{\pi_\theta}(\hat{s}, a) \nabla_\theta \log \pi_\theta(a|\hat{s})$, over $a \sim \pi_\theta(\hat{s})$. Using these, the gradient estimate is obtained via (13). Note that $\mathcal{O}((1-\gamma)^{-1}A)$ samples are required on average to run one iteration of Algorithm 1.

Although our sampling process is similar to those used in earlier works (Liu et al., 2020; Mondal & Aggarwal, 2024b), there are some crucial differences. In particular, the incorporation of the expectation over $a \sim \pi_\theta(\hat{s})$ to obtain $\hat{F}(\theta)$ and $\hat{H}_\tau(\theta, \lambda)$ is a new addition that was not present in the previous papers. Such a seemingly insignificant change has major ramifications for the global convergence result, as explained later in the paper (see the discussion following Lemma 7). Lemma 4 (stated below) demonstrates that the estimates obtained from Algorithm 1 are unbiased.

---

[3]The symbol $\mathrm{Geo}(\cdot)$ denotes the geometric distribution.

**Lemma 4.** *The estimates obtained from Algorithm 1 are unbiased. Formally, for arbitrary $\omega, \theta, \lambda, \tau$, we get*

$$\mathbf{E}\left[\hat{J}_c(\theta)\big|\theta\right] = J_c(\theta), \ \text{and} \tag{19}$$

$$\mathbf{E}\left[\hat{\nabla}_\omega \mathcal{E}^\tau_{\nu^{\pi_\theta}}(\omega, \theta, \lambda)\big|\omega, \theta, \lambda\right] = \nabla_\omega \mathcal{E}^\tau_{\nu^{\pi_\theta}}(\omega, \theta, \lambda) \tag{20}$$

---

**Algorithm 2** Primal-Dual Regularized Accelerated Natural Policy Gradient (PDR-ANPG)

---

1: **Input:** Parameters $(\theta_0, \lambda_0, \tau, K, H)$, Distribution $\rho$, Learning Rates $\eta, \alpha, \beta, \xi, \delta$

2:                                                        ▷ Outer Loop

3: **for** $k \in \{0, \cdots, K-1\}$ **do**

4:      $\mathbf{x}_0, \mathbf{v}_0 \leftarrow \mathbf{0}$

5:                                                        ▷ Inner Loop

6:      **for** $h \in \{0, \cdots, H-1\}$ **do**

7:                                                  ▷ Accelerated SGD

$$\mathbf{y}_h \leftarrow \alpha \mathbf{x}_h + (1-\alpha)\mathbf{v}_h \tag{21}$$

8:          $\hat{G}_h \leftarrow \hat{\nabla}_\omega \mathcal{E}^\tau_{\nu^{\pi_{\theta_k}}_\rho}(\omega, \theta_k, \lambda_k)\big|_{\omega = \mathbf{y}_h}$ (Algo. 1)

$$\mathbf{x}_{h+1} \leftarrow \mathbf{y}_h - \delta \hat{G}_h \tag{22}$$

$$\mathbf{z}_h \leftarrow \beta \mathbf{y}_h + (1-\beta)\mathbf{v}_h \tag{23}$$

$$\mathbf{v}_{h+1} \leftarrow \mathbf{z}_h - \xi \hat{G}_h \tag{24}$$

9:      Tail Averaging:

$$\omega_k \leftarrow \frac{2}{H} \sum_{\frac{H}{2} < h \leq H} \mathbf{x}_h \tag{25}$$

10:      Obtain $\hat{J}_{c,\rho}(\theta_k)$ via Algorithm 1.

11:      Parameter Updates:

$$\theta_{k+1} \leftarrow \theta_k + \eta \omega_k \tag{26}$$

$$\lambda_{k+1} \leftarrow \mathcal{P}_\Lambda[\lambda_k(1-\eta\tau) - \eta\hat{J}_c(\theta_k)] \tag{27}$$

12: **Output:** $\{(\theta_k, \lambda_k)\}_{k=1}^K$

---

The estimates obtained from Algorithm 1 are used in Algorithm 2, which runs in nested loops. The *outer loop* executes $K$ number of times and, at the $k$th instant, updates the current primal and dual parameters $\theta_k, \lambda_k$ via equation 26 and equation 27 respectively. The estimate $\hat{J}_c(\theta_k)$ is obtained via Algorithm 1. On the other hand, the direction $\omega_k$, (which is used in the primal update), is computed in the inner loop by iteratively minimizing the function $\mathcal{E}^\tau_{\nu^{\pi_{\theta_k}}_\rho}(\cdot, \theta_k, \lambda_k)$ in $H$-steps using an Accelerated Stochastic Gradient Descent (ASGD) routine as stated in Jain et al. (2018). Each ASGD iteration comprises the updates equation 21 − equation 24, followed by a tail averaging step equation 25 where the gradient estimates $\hat{G}_h$'s are yielded by Algorithm 1 and $(\alpha, \beta, \xi, \delta)$ are appropriate learning parameters.

## 5 Last-Iterate Convergence Analysis

This section shows the last-iterate convergence properties of Algorithm 2. Before presenting the technical results, we will outline a few key assumptions required for the analysis.

**Assumption 2.** *The policy is differentiable, and the score function is bounded and Lipschitz continuous, i.e., $\forall \theta, \theta_1, \theta_2 \in \mathbb{R}^d$ and $\forall(s,a)$,*

$$\|\nabla_\theta \log \pi_\theta(a|s)\| \le G,$$
$$\|\nabla_\theta \log \pi_{\theta_1}(a|s) - \nabla_\theta \log \pi_{\theta_2}(a|s)\| \le B\|\theta_1 - \theta_2\|$$

*where $B, G > 0$ are constants.*

Assumption 2 is common in the convergence analysis of policy gradient-based algorithms, e.g., see (Fatkhullin et al., 2023; Zhang et al., 2021b; Liu et al., 2020). The above assumption essentially ensures that the policy does not change rapidly as a result of a slight nudge in the $\theta$ parameter. In the absence of such a guarantee, it would be very difficult to arrive at a stationary $\theta$ point. This assumption is obeyed by many policy classes, such as neural networks with bounded parameters. The result stated below can be obtained by combining the above assumption with Lemma 2 and 3.

**Lemma 5.** *The following inequality holds $\forall \theta, \lambda, \tau$ where $L_{\tau,\lambda}^2$ is defined in (16).*

$$\|\nabla_\theta \mathcal{L}_\tau(\theta, \lambda)\|^2 \le (1-\gamma)^{-2} G^2 L_{\tau,\lambda}^2 \tag{28}$$

Lemma 5 dictates that the norm of the gradient of the Lagrange function can be bounded above by some function of problem-dependent parameters. This result will be useful in the convergence analysis in the forthcoming section.

**Assumption 3.** *The approximation error function defined in (17) satisfies the following*

$$\mathcal{E}_{\nu^{\pi_\tau^*}}^\tau(\omega_\tau^*(\theta, \lambda), \theta, \lambda) \le \frac{1}{2}\epsilon_{\text{bias}} \tag{29}$$

*where $\theta \in \mathbb{R}^d$, $\lambda \in \Lambda$, $\tau \in [0,1]$, and $\omega_\tau^*(\theta, \lambda)$ defines the NPG. The LHS of (29) is defined as the transferred compatibility approximation error.*

The term $\epsilon_{\text{bias}}$ utilized in Assumption 3 can be interpreted as the expressivity error of the parameterized policy class. This term becomes zero for complete policy classes, e.g., softmax tabular policies Agarwal et al. (2021). One can also exhibit $\epsilon_{\text{bias}} = 0$ for linear MDPs (Yang & Wang, 2019; Jin et al., 2020). Moreover, for certain rich policy classes (e.g., wide neural networks), $\epsilon_{\text{bias}}$ turns out to be small (Wang et al., 2020).

**Assumption 4.** *There exists $\mu_F > 0$ such that $F(\theta) - \mu_F I_d$ is positive semidefinite, i.e., $F(\theta) \succeq \mu_F I_d$, where $F(\theta)$ is the Fisher matrix given in equation 12, $I_d$ is a $d \times d$ the identity matrix, and $\theta \in \mathbb{R}^d$.*

Assumption 4 dictates that the Fisher matrix $F(\theta)$ is positive definite, and its eigenvalues are lower bounded by some positive constant $\mu_F$. It is also easy to check that $F(\theta)$ is the Hessian of the function $\mathcal{E}_{\nu^{\pi_\theta}}^\tau(\cdot, \theta, \lambda)$. Assumption 4, thus, implies that the function $\mathcal{E}_{\nu^{\pi_\theta}}^\tau(\cdot, \theta, \lambda)$ is $\mu_F$-strongly convex, and its minimizer $\omega_\tau^*(\theta, \lambda)$ (the NPG) is unique. It is well known in the optimization literature that we can approach the global minimum of a strongly convex function exponentially fast, given access to the exact gradient. Lemma 8 will exhibit that such insights are crucial in achieving the desired convergence rate. A policy class that satisfies Assumption 4 is called a Fisher Non-Degenerate class. Such a restriction on the structure of the policy class is common across the literature (Fatkhullin et al., 2023; Zhang et al., 2021a; Bai et al., 2024). A detailed discussion on the class of policies that obey the above assumption is available in (Fatkhullin et al., 2023). It is worth noting that softmax-based policies may not satisfy the above assumption if they are close to being deterministic. However, if the softmax parameters are restricted within a finite interval, then they obey this assumption (Mondal et al., 2023).

## 5.1 Analysis of the Outer Loop

Let $(\theta_k, \lambda_k)$ denotes the parameters generated by Algorithm 2 after $k$ iterations of the outer loop. Let

$$\Phi_k^\tau \triangleq \mathbf{E}\left[\text{KL}_k^\tau\right] + \frac{1}{2}\mathbf{E}\left[(\lambda_\tau^* - \lambda_k)^2\right], \text{ where}$$
$$\text{KL}_k^\tau \triangleq \sum_{s \in \mathcal{S}} d^{\pi_\tau^*}(s)\text{KL}\left(\pi_\tau^*(\cdot|s)||\pi_{\theta_k}(\cdot|s)\right) \tag{30}$$

where $\mathrm{KL}(\cdot||\cdot)$ is the KL-divergence, and $\pi_\tau^*, \lambda_\tau^*$ are defined in equation 9. The expectations are computed over the distributions of $(\theta_k, \lambda_k)$. Intuitively, the term $\Phi_k^\tau$ captures the distance between the solution provided by Algorithm 2 and the optimal regularized solution. Lemma 6 (stated below) establishes a recursive inequality obeyed by $\Phi_k^\tau$.

**Lemma 6.** *Let the parameters $\{\theta_k, \lambda_k\}_{k=0}^K$ be updated via equation 26, equation 27, and $\tau \in [0,1]$. The following relation holds under assumptions 2 and 3, $\forall k \in \{0, 1, \cdots, K-1\}$.*

$$
\begin{aligned}
\Phi_{k+1}^\tau \leq &(1 - \eta\tau)\Phi_k^\tau + \eta\sqrt{\epsilon_{\mathrm{bias}}} \\
&+ \eta G \mathbf{E} \left\| \mathbf{E}\left[\omega_k | \theta_k, \lambda_k\right] - \omega_{k,\tau}^* \right\| + \frac{B\eta^2}{2} \mathbf{E}\|\omega_k\|^2 + \eta^2 \left[ \frac{2}{(1-\gamma)^2} + \tau^2 \lambda_{\max}^2 \right]
\end{aligned}
\tag{31}
$$

*where $\omega_{k,\tau}^* \triangleq \omega_\tau^*(\theta_k, \lambda_k)$ is the exact NPG at $(\theta_k, \lambda_k)$.*

Note the term $\epsilon_{\mathrm{bias}}$ in (31). In Theorem 1, we show that its presence indicates that the optimality gap cannot be forced to zero due to the incompleteness of parameterized policies. Note that the term $\mathbf{E}\|\mathbf{E}[\omega_k|\theta_k, \lambda_k] - \omega_{k,\tau}^*\|$ is the expected bias of the NPG estimator. On the other hand,

$$
\begin{aligned}
\mathbf{E}\|\omega_k\|^2 &\leq 2\mathbf{E}\|\omega_k - \omega_{k,\tau}^*\|^2 + 2\|F(\theta_k)^\dagger \nabla_\theta \mathcal{L}_\tau(\theta_k, \lambda_k)\|^2 \\
&\overset{(a)}{\leq} 2\mathbf{E}\|\omega_k - \omega_{k,\tau}^*\|^2 + 2\mu_F^{-2}\|\nabla_\theta \mathcal{L}_\tau(\theta_k, \lambda_k)\|^2
\end{aligned}
\tag{32}
$$

where $(a)$ follows from Assumption 4. Note that the second term in equation 32 can be bounded by Lemma 5. Next section provides bounds for both the first-order term $\mathbf{E}\|\mathbf{E}[\omega_k|\theta_k, \lambda_k] - \omega_{k,\tau}^*\|$ and the second-order term $\mathbf{E}\|\omega_k - \omega_{k,\tau}^*\|^2$ used in equation 31 and equation 32 respectively.

## 5.2 Analysis of the Inner Loop

We start with a statistical characterization of the noise of the gradient estimator given by Algorithm 1.

**Lemma 7.** *The estimate $\hat{\nabla}_\omega \mathcal{E}_{\nu^{\pi_\theta}}^\tau(\omega, \theta, \lambda)$ given by Algorithm 1 obeys the following semidefinite inequality if assumptions 2 and 4 holds.*

$$
\mathbf{E}\left[ \hat{\nabla}_\omega \mathcal{E}_{\nu^{\pi_\theta}}^\tau(\omega_\tau^*(\theta, \lambda), \theta, \lambda) \otimes \hat{\nabla}_\omega \mathcal{E}_{\nu^{\pi_\theta}}^\tau(\omega_\tau^*(\theta, \lambda), \theta, \lambda) \right] \preceq \sigma_{\tau,\lambda}^2 F(\theta)
\tag{33}
$$

*where $\theta, \lambda, \tau$ are arbitrary, $F(\theta)$ is the Fisher matrix defined in (12), and $\sigma_{\tau,\lambda}^2$ is given as follows.*

$$
\sigma_{\tau,\lambda}^2 = \frac{48}{(1-\gamma)^4}\left[1 + \lambda^2 + 4Ae^{-2}\tau^2\right] + \frac{2G^4 L_{\tau,\lambda}^2}{\mu_F^2(1-\gamma)^2}
\tag{34}
$$

*where $L_{\tau,\lambda}^2$ is defined in (16).*

The term $\sigma_{\tau,\lambda}^2$ can be interpreted as the scaled noise variance of the gradient estimator. Hence, if we access the exact gradient oracle, $\sigma_{\tau,\lambda}^2$ will be zero. This observation will turn out to be useful later. Before describing that result, we would like to present some insights about the connection between Lemma 7 and the gradient estimation procedure (Algorithm 1). To understand the crux of the proof of Lemma 7, define the following.

$$
\hat{\zeta}_{\theta,\lambda}^\tau(s,a) \triangleq \nabla_\theta \log \pi_\theta(a|s) \cdot \omega_\tau^*(\theta, \lambda) - \frac{1}{1-\gamma} \hat{A}_{r+\lambda c + \tau\psi_\theta}^{\pi_\theta}(s,a), \ \forall(s,a)
\tag{35}
$$

where $\hat{A}_g^{\pi_\theta}(s,a)$, $g = r + \lambda c + \tau\psi_\theta$ is the advantage estimate given by Algorithm 1 for a given $(s,a)$. Following (13), we can conclude that

$$
\hat{\nabla}_\omega \mathcal{E}_{\nu^{\pi_\theta}}^\tau(\omega_\tau^*(\theta, \lambda), \theta, \lambda) = \mathbf{E}_{a \sim \pi_\theta(\hat{s})}\left[ \hat{\zeta}_{\theta,\lambda}^\tau(\hat{s}, a) \nabla_\theta \log \pi_\theta(a|\hat{s}) \right]
\tag{36}
$$

where one can demonstrate that $\hat{s} \sim d^{\pi_\theta}$. Let $\hat{\mathbf{E}}^\theta_{s,a}$ be the expectation over the distribution of $\pi_\theta$-induced trajectories that are used in evaluating the advantage term $\hat{A}^{\pi_\theta}_{r+\lambda c+\tau\psi_\theta}(s,a)$. Also, let $\hat{\mathbf{E}}^\theta_s$ be the expectation computed over the joint distribution of $\pi_\theta$-induced trajectories that are used in estimating $\{\hat{A}^{\pi_\theta}_{r+\lambda c+\tau\psi_\theta}(s,a)\}_{a\in\mathcal{A}}$. Following the sampling process (Algorithm 1), one gets

$$\mathbf{E}_{a\sim\pi_\theta(s)}\hat{\mathbf{E}}^\theta_{s,a}\left[\left(\hat{\zeta}^\tau_{\theta,\lambda}(s,a)\right)^2\right] \leq \sigma^2_{\tau,\lambda}, \ \forall s \tag{37}$$

Note that the LHS of (33) obeys the following inequality.

$$\text{LHS} = \mathbf{E}_{s\sim d^{\pi_\theta}}\hat{\mathbf{E}}^\theta_s\left[\mathbf{E}_{a\sim\pi_\theta(s)}\left[\hat{\zeta}^\tau_{\theta,\lambda}(s,a)\nabla_\theta\log\pi_\theta(a|s)\right] \otimes \mathbf{E}_{a\sim\pi_\theta(s)}\left[\hat{\zeta}^\tau_{\theta,\lambda}(s,a)\nabla_\theta\log\pi_\theta(a|s)\right]\right]$$

$$\overset{(a)}{\preceq} \mathbf{E}_{s\sim d^{\pi_\theta}}\left[\mathbf{E}_{a\sim\pi_\theta(s)}\left[\hat{\mathbf{E}}^\theta_{s,a}\left[\left(\hat{\zeta}^\tau_{\theta,\lambda}(s,a)\right)^2\right]\right] \times \mathbf{E}_{a\sim\pi_\theta(s)}\left[\nabla_\theta\log\pi_\theta(a|s)\otimes\nabla_\theta\log\pi_\theta(a|s)\right]\right]$$

where $(a)$ is a result of the Cauchy-Schwarz inequality and interchange of expectation operations. Now, (33) can be proven by using (37) in the above relation. Notice the importance of the expectation over $a\sim\pi_\theta(\hat{s})$ in the sampling procedure. Had the gradient been estimated without this step (which is typically done in the literature, see our earlier discussion), the LHS of (33) would have been,

$$\text{LHS} = \mathbf{E}_{s\sim d^{\pi_\theta}}\mathbf{E}_{a\sim\pi_\theta(s)}\left[\underbrace{\hat{\mathbf{E}}^\theta_{s,a}\left[\left(\hat{\zeta}^\tau_{\theta,\lambda}(s,a)\right)^2\right]}_{T_0(s,a)}\times\nabla_\theta\log\pi_\theta(a|s)\otimes\nabla_\theta\log\pi_\theta(a|s)\right]$$

which is impossible to bound since $T_0(s,a)$ is not uniformly bounded $\forall(s,a)$ due to the presence of the term $\psi_\theta$ originating from the entropy regularizer. In simple terms, due to our unconventional sampling procedure, the LHS of (33) takes the form $\mathbf{E}[XY]\otimes\mathbf{E}[XY]$, where $X = \hat{\zeta}^\tau_{\theta,\lambda}(\hat{s},a)$, and $Y = \nabla_\theta\log\pi_\theta(a|\hat{s})$. We can now use the Cauchy-Schwarz inequality to deduce that $\mathbf{E}[XY]\otimes\mathbf{E}[XY] \preceq \mathbf{E}[X^2]\mathbf{E}[Y\otimes Y]$. This is useful since it allows us to apply the bound on $\mathbf{E}[X^2]$ given in (37). However, without our sampling procedure, the LHS of (33) takes the form $\mathbf{E}[X^2Y\otimes Y]$, which cannot be directly bounded because the random variable $X$, due to its very definition, is not almost surely bounded. Moreover, the inequality $\mathbf{E}[X^2Y\otimes Y] \preceq \mathbf{E}[X^2]\mathbf{E}[Y\otimes Y]$ is not true in general. This negates the possibility of applying the bound of $\mathbf{E}[X^2]$.

**Lemma 8.** *If assumptions 2, 4 are true, the following holds $\forall k \in \{0, \cdots, K-1\}$ with $\alpha = \frac{3\sqrt{5}G^2}{\mu_F+3\sqrt{5}G^2}$, $\beta = \frac{\mu_F}{9G^2}$, $\xi = \frac{1}{3\sqrt{5}G^2}$, and $\delta = \frac{1}{5G^2}$ if the inner loop length of Algorithm 2 obeys $H > \max\left\{1, \bar{C}\frac{G^2}{\mu_F}\log\left(\sqrt{d}\frac{G^2}{\mu_F}\right)\right\}$ for some universal constant, $\bar{C}$.*

$$\mathbf{E}\|\omega_k - \omega^*_{k,\tau}\|^2 \leq 22\frac{\sigma^2_{\tau,\lambda_{\max}}d}{\mu_F} + \frac{CL^2_{\tau,\lambda_{\max}}}{\mu_F(1-\gamma)^2}, \tag{38}$$

$$\mathbf{E}\|\mathbf{E}\left[\omega_k\big|\theta_k,\lambda_k\right] - \omega^*_{k,\tau}\| \leq \sqrt{C}\exp\left(-\frac{\mu_F}{40G^2}H\right)\left[\frac{L_{\tau,\lambda_{\max}}}{\sqrt{\mu_F}(1-\gamma)}\right] \tag{39}$$

*where $C$ is a constant, $\omega_k$ is defined in (25), $\omega^*_{k,\tau}$ is the exact NPG, and $\sigma^2_{\tau,\lambda_{\max}}$ is given by (34).*

Inequality (38) can be established utilizing Corollary 2 of (Jain et al., 2018) in our framework and simplifying the upper bound. However, (39) follows from the same result applied to the scenario when an exact gradient oracle is available. To understand the reason, note that we can write the following equations using conditional expectation on both sides of the update equations $(21)-(24)$, and Lemma 4.

$$\begin{aligned}
\bar{\mathbf{y}}_h &= \alpha\bar{\mathbf{x}}_h + (1-\alpha)\bar{\mathbf{v}}_h,\\
\bar{\mathbf{x}}_{h+1} &= \bar{\mathbf{y}}_h - \delta\nabla_\omega\mathcal{E}^{\tau_{\pi_{\theta_k}}}_\nu(\omega,\theta_k,\lambda_k)\big|_{\omega=\bar{\mathbf{y}}_h}\\
\bar{\mathbf{z}}_h &= \beta\bar{\mathbf{y}}_h + (1-\beta)\bar{\mathbf{v}}_h,\\
\bar{\mathbf{v}}_{h+1} &= \bar{\mathbf{z}}_h - \xi\nabla_\omega\mathcal{E}^{\tau_{\pi_{\theta_k}}}_{\nu_\rho}(\omega,\theta_k,\lambda_k)\big|_{\omega=\bar{\mathbf{y}}_h}
\end{aligned} \tag{40}$$

where $\bar{l}_h = \mathbf{E}[l_h|\theta_k, \lambda_k]$, $\forall h \in \{0, \cdots, H-1\}$, $l \in \{\mathbf{v}, \mathbf{x}, \mathbf{y}, \mathbf{z}\}$. Invoking (25), we get,

$$\bar{\omega}_k \triangleq \mathbf{E}\left[\omega_k|\theta_k, \lambda_k\right] = \frac{2}{H} \sum_{\frac{H}{2} < h \leq H} \bar{\mathbf{x}}_h \tag{41}$$

Note that equation 40 and equation 41 emulate the steps of an ASGD process with exact (deterministic) gradients. We can now combine Lemma 6, equation 32, and Lemma 8 to arrive at the following corollary.

**Corollary 1.** *Let $\{\theta_k, \lambda_k\}_{k=0}^K$ be obtained via (26), (27), and $\tau \in [0,1]$. The following holds $\forall k \in \{1, \cdots, K\}$ if assumptions 2−4 are true, $(\alpha, \beta, \xi, \delta)$ are the same as in Lemma 8, $\eta\tau < 1$, and $H$ is sufficiently large.*

$$\Phi_k^\tau \leq \exp(-\eta\tau k)\Phi_0^\tau + \frac{1}{\tau}\sqrt{\epsilon_{\text{bias}}} + \frac{1}{\tau}R_0 \exp\left(-\frac{\mu_F}{40G^2}H\right) + \frac{\eta}{\tau}(R_1 + R_2) \tag{42}$$

*where $R_0, R_1, R_2$ are as follows.*

$$R_0 \triangleq \frac{G\sqrt{C}L_{1,\lambda_{\max}}}{\sqrt{\mu_F}(1-\gamma)}, \;\; R_1 \triangleq \frac{2}{(1-\gamma)^2} + \lambda_{\max}^2, \;\; R_2 \triangleq \frac{B}{\mu_F}\left[22\sigma_{1,\lambda_{\max}}^2 \mathrm{d} + \frac{C + \mu_F^{-1}G^2}{(1-\gamma)^2}L_{1,\lambda_{\max}}^2\right]$$

## 5.3 Optimality Gap and Constraint Violation

This section discusses how the optimality gap and constraint violation rates can be extracted from the convergence result of $\Phi_k^\tau$ stated in Corollary 1.

**Theorem 1.** *Let $\{(\theta_k, \lambda_k)\}_{k=1}^K$ be the parameters generated by Algorithm 2 starting from $(\theta_0, \lambda_0)$. Assume that assumptions 1−4 hold, the parameters $(\alpha, \beta, \xi, \delta)$ are given by Lemma 8, $\lambda_{\max} = 4/[(1-\gamma)c_{\text{slat}}]$, $\epsilon_{\text{bias}} < 1$, and $\epsilon < 1$ is sufficiently small. The relations stated below hold if $\tau = \max\{\epsilon, (\epsilon_{\text{bias}})^{\frac{1}{6}}\}$, $K = 2\epsilon^{-2}\tau^{-2}$, $\eta = \epsilon^2\tau$, and $H = 40G^2\mu_F^{-1}\log\left(\tau^{-1}\epsilon^{-2}\right)$.*

$$J_r^{\pi^*} - \mathbf{E}\left[J_r^{\pi_{\theta_K}}\right] = \mathcal{O}(\epsilon + (\epsilon_{\text{bias}})^{\frac{1}{6}}), \quad and$$
$$\mathbf{E}\left[-J_c^{\pi_{\theta_K}}\right] = \mathcal{O}(\epsilon + (\epsilon_{\text{bias}})^{\frac{1}{6}}) \tag{43}$$

Theorem 1 essentially states that for any parameterized policy class with a transferred compatibility approximation error $\epsilon_{\text{bias}}$, we can substitute $H = \tilde{O}(1)$, $K = \mathcal{O}(\epsilon^{-2}\min\{\epsilon^{-2}, (\epsilon_{\text{bias}})^{-\frac{1}{3}}\})$ to ensure the (last-iterate) optimality gap and the constrained violation to be $\mathcal{O}(\epsilon + (\epsilon_{\text{bias}})^{\frac{1}{6}})$. This makes the overall sample complexity $\tilde{\mathcal{O}}(\epsilon^{-2}\min\{\epsilon^{-2}, (\epsilon_{\text{bias}})^{-\frac{1}{3}}\})$. The following remarks are worth highlighting.

Firstly, if $\epsilon_{\text{bias}} > 0$ and independent of $\epsilon$, then the generated policies of Algorithm 2 cannot be guaranteed to be arbitrarily close to the optimal policy. This agrees with the existing literature on general parameterization (Liu et al., 2020; Fatkhullin et al., 2023; Mondal & Aggarwal, 2024b). In this scenario, we obtain the optimal $\tilde{\mathcal{O}}(\epsilon^{-2})$ sample complexity for $\epsilon < (\epsilon_{\text{bias}})^{\frac{1}{6}}$. On the other hand, if $\epsilon_{\text{bias}} = 0$, i.e., the policy class is complete, the optimality gap and constraint violation can be made $\mathcal{O}(\epsilon)$ at the cost of $\tilde{\mathcal{O}}(\epsilon^{-4})$ sample complexity.

Secondly, some works take $\epsilon_{\text{bias}}$ to be dependent on $\epsilon$. For example, Ding et al. (2024) assumes $\epsilon_{\text{bias}} = \mathcal{O}(\epsilon^8)$ for log-linear classes to get a sample complexity of $\tilde{\mathcal{O}}(\epsilon^{-6})$ while enjoying $\mathcal{O}(\epsilon)$ optimality gap and constraint violation. If, following a similar path, one selects $\epsilon_{\text{bias}} = \Theta(\epsilon^6)$ in Theorem 1, the sample complexity becomes $\tilde{\mathcal{O}}(\epsilon^{-4})$ while the optimality gap and the constraint violation change to $\mathcal{O}(\epsilon)$.

Thirdly, although our result does not establish zero constraint violation, it can be shown using a "conservative constraint" trick. The main idea is to apply a conservative constraint $c' = c - (1-\gamma)\delta'$ where $\delta'$ is a tunable parameter. Observe that $J_{c'}^\pi = J_c^\pi - \delta'$ for any arbitrary policy $\pi$. Hence, once a $\mathcal{O}(\epsilon)$ constraint violation is proven for $c'$ (utilizing the same algorithm described in our paper), we can judiciously choose $\delta'$ to prove zero constraint violation for $c$ (see details in Ding et al. (2024); Bai et al. (2023)).

Fourthly, the choice of some of our parameters is dependent on $\epsilon_{\text{bias}}$. We acknowledge that the knowledge of $\epsilon_{\text{bias}}$ may not be available for some parameterized policy classes. However, it might be possible to obtain an upper bound, $\bar{\epsilon}_{\text{bias}}$, of the desired quantity. For example, Wang et al. (2020) shows that for a two-layer neural parameterization of width $m$, the function approximation error is $\mathcal{O}(m^{-1/8})$ (see Theorem A.4 and

the subsequent discussion). Our algorithm still works if this bound is used as a proxy of $\epsilon_{\text{bias}}$. In this case, the convergence error and the sampling complexity will be functions of $\bar{\epsilon}_{\text{bias}}$, instead of $\epsilon_{\text{bias}}$.

Finally, notice the importance of the observation that the NPG bias $\mathbf{E}[\omega_k|\theta_k, \lambda_k] - \omega_{k,\tau}^*$ is exactly the convergence error of an ASGD routine with exact gradients (see Lemma 8 and its subsequent discussion). Without this insight, the expected bias would be $\mathcal{O}(H^{-\frac{1}{2}})$, instead of its current bound $\exp(-\Theta(H))$ which would degrade the overall sample complexity by many folds.

## 6 Conclusions

We present an algorithm for the general parametrized CMDP that ensures $\mathcal{O}(\epsilon + \epsilon_{\text{bias}}^{\frac{1}{6}})$ last-iterate optimality gap and the same constraint violation with $\tilde{\mathcal{O}}(\epsilon^{-2} \min\{\epsilon^{-2}, (\epsilon_{\text{bias}})^{-\frac{1}{3}}\})$ sample complexity. Here $\epsilon_{\text{bias}}$ denotes the transferred compatibility approximation error of the underlying policies. Future work includes improving the sample complexity to $\tilde{\mathcal{O}}(\epsilon^{-2})$ across all parameterized policy classes, extending our ideas to general utility CMDPs, and proving a high-probability sample complexity bound, which would strengthen our expectation-based result. Additionally, extending the results to multiple constraints would be an interesting problem to tackle. Finally, our current results apply to finite action spaces. Extending it to infinite action spaces would be another major hurdle to overcome in the future.

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

# A    Proof of Lemma 2

Fix a tuple $(\theta, \lambda, \tau)$. The following proof assumes that the policy and all relevant value functions are differentiable w. r. t. $\theta$. We also assume the state space, $\mathcal{S}$, to be finite to avoid technicalities associated with infinite space. Note that, $\forall s \in \mathcal{S}$, we have,

$$
\begin{aligned}
\nabla_\theta V^{\pi_\theta}_{r+\lambda c+\tau\psi_\theta}(s) &= \nabla_\theta \sum_{a\in\mathcal{A}} \pi_\theta(a|s) Q^{\pi_\theta}_{r+\lambda c+\tau\psi_\theta}(s,a) \\
&= \sum_{a\in\mathcal{A}} Q^{\pi_\theta}_{r+\lambda c+\tau\psi_\theta}(s,a)\nabla_\theta\pi_\theta(a|s) + \underbrace{\sum_{a\in\mathcal{A}}\pi_\theta(a|s)\nabla_\theta Q^{\pi_\theta}(s,a)}_{T_0}
\end{aligned}
\tag{44}
$$

Applying the Bellman equation, one obtains,

$$
\begin{aligned}
T_0 &= \sum_{a\in\mathcal{A}} \pi_\theta(a|s)\nabla_\theta \left\{ r(s,a) + \lambda c(s,a) - \tau\log\pi_\theta(a|s) + \gamma\sum_{s'\in\mathcal{S}} P(s'|s,a) V^{\pi_\theta}_{r+\lambda c+\tau\psi_\theta}(s') \right\} \\
&= -\tau \underbrace{\sum_{a\in\mathcal{A}} \pi_\theta(a|s)\nabla_\theta\log\pi_\theta(a|s)}_{=0} + \gamma\sum_{s'\in\mathcal{S}}\sum_{a\in\mathcal{A}} \pi_\theta(a|s)P(s'|s,a)\nabla_\theta V^{\pi_\theta}_{r+\lambda c+\tau\psi_\theta}(s') \\
&= \gamma\sum_{s'\in\mathcal{S}} \Pr\left(s_1=s'\middle|s_0=s,\pi_\theta\right)\nabla_\theta V^{\pi_\theta}_{r+\lambda c+\tau\psi_\theta}(s')
\end{aligned}
\tag{45}
$$

Combining the above results, we obtain,

$$
\begin{aligned}
&\nabla_\theta V^{\pi_\theta}_{r+\lambda c+\tau\psi_\theta}(s) \\
&= \sum_{a\in\mathcal{A}} Q^{\pi_\theta}_{r+\lambda c+\tau\psi_\theta}(s,a)\nabla_\theta\pi_\theta(a|s) + \gamma\sum_{s'\in\mathcal{S}}\Pr\left(s_1=s'\middle|s_0=s,\pi_\theta\right)\nabla_\theta V^{\pi_\theta}_{r+\lambda c+\tau\psi_\theta}(s') \\
&\overset{(a)}{=} \sum_{a\in\mathcal{A}} A^{\pi_\theta}_{r+\lambda c+\tau\psi_\theta}(s,a)\nabla_\theta\pi_\theta(a|s) + \gamma\sum_{s'\in\mathcal{S}}\Pr\left(s_1=s'\middle|s_0=s,\pi_\theta\right)\nabla_\theta V^{\pi_\theta}_{r+\lambda c+\tau\psi_\theta}(s') \\
&\overset{(b)}{=} \sum_{s'\in\mathcal{S}}\sum_{t=0}^{T-1}\gamma^t\Pr(s_t=s'|s_0=s,\pi_\theta)\sum_{a\in\mathcal{A}} A^{\pi_\theta}_{r+\lambda c+\tau\psi_\theta}(s,a)\nabla_\theta\pi_\theta(a|s) \\
&\qquad\qquad\qquad\qquad + \gamma^T\sum_{s'\in\mathcal{S}}\Pr(s_T=s'|s_0=s,\pi_\theta)\nabla_\theta V^{\pi_\theta}(s')
\end{aligned}
\tag{46}
$$

where $(a)$ uses the fact that $\sum_a \nabla_\theta\pi_\theta(a|s) = 0$, $\forall s \in \mathcal{S}$, and $(b)$ is obtained by repeatedly applying the recursion. Notice that, for finite $\mathcal{S}$, we have the following limit.

$$
\gamma^T\sum_{s'\in\mathcal{S}}\Pr(s_T=s'|s_0=s,\pi_\theta)\nabla_\theta V^{\pi_\theta}(s') \to \mathbf{0} \text{ as } T\to\infty
\tag{47}
$$

Finally, applying $T\to\infty$ to equation 46, we obtain the following,

$$
\begin{aligned}
\nabla_\theta\mathcal{L}_\tau(\theta,\lambda) &= \mathbf{E}_{s\sim\rho}\left[\nabla_\theta V^{\pi_\theta}_{r+\lambda c+\tau\psi_\theta}(s)\right] \\
&= \sum_{s'\in\mathcal{S}}\sum_{t=0}^{\infty}\gamma^t\Pr(s_t=s'|s_0\sim\rho,\pi_\theta)\sum_{a\in\mathcal{A}} Q^{\pi_\theta}_{r+\lambda c+\tau\psi_\theta}(s,a)\nabla_\theta\pi_\theta(a|s) \\
&= \frac{1}{1-\gamma}\mathbf{E}_{(s,a)\sim\nu^{\pi_\theta}}\left[A^{\pi_\theta}_{r+\lambda c+\tau\psi_\theta}(s,a)\nabla_\theta\log\pi_\theta(a|s)\right]
\end{aligned}
\tag{48}
$$

This establishes the lemma. It is worth pointing out that, although our proof is similar to that of the standard policy gradient theorem Sutton et al. (1999), there are some differences. In particular, the standard proof assumes the reward/utility function to be independent of $\theta$. This does not hold in our case due to the presence of $\psi_\theta$.

## B  Proof of Lemma 3

Fix a tuple $(\theta, \lambda, \tau, s)$. Since $r$ and $c$ are absolutely bounded in $[0, 1]$, we get,

$$\left|A^{\pi_\theta}_{r+\lambda c}(s, a)\right| \leq \left|Q^{\pi_\theta}_{r+\lambda c}(s, a)\right| + \left|V^{\pi_\theta}_{r+\lambda c}(s)\right| \leq \frac{2(1+\lambda)}{1-\gamma}, \ \forall a \tag{49}$$

Furthermore, observe that,

$$0 \leq V^{\pi_\theta}_{\psi_\theta}(s) = \mathbf{E}_{\pi_\theta}\left[\sum_{t=0}^\infty -\gamma^t \log \pi_\theta(a_t|s_t)\Big|s_0 = s\right]$$
$$= -\frac{1}{1-\gamma}\sum_{s' \in \mathcal{S}} d^{\pi_\theta}_s(s')\sum_{a \in \mathcal{A}} \pi_\theta(a|s')\log \pi_\theta(a|s') \overset{(a)}{\leq} \frac{1}{1-\gamma}\log A \tag{50}$$

where $d^{\pi_\theta}_s$ is defined similarly as $d^{\pi_\theta}$ given in equation 2 but with the conditional event $s_0 \sim \rho$ being replaced by $s_0 = s$. Inequality $(a)$ follows from the concavity of the log function and the fact that $d^{\pi_\theta}_s \in \Delta(\mathcal{S})$. We deduce the following.

$$\mathbf{E}_{a \sim \pi_\theta(s)}\left[\left|A^{\pi_\theta}_{\psi_\theta}(s, a)\right|^2\right]$$
$$\leq 2\mathbf{E}_{a \sim \pi_\theta(s)}\left[\left|Q^{\pi_\theta}_{\psi_\theta}(s, a)\right|^2\right] + 2\left[V^{\pi_\theta}_{\psi_\theta}(s)\right]^2$$
$$\overset{(a)}{\leq} 2\mathbf{E}_{a \sim \pi_\theta(s)}\left[\left|-\log \pi_\theta(a|s) + \gamma \mathbf{E}_{s' \sim P(s,a)}\left[V^{\pi_\theta}_{\psi_\theta}(s')\right]\right|^2\right] + \frac{2(\log A)^2}{(1-\gamma)^2} \tag{51}$$
$$\overset{(b)}{\leq} 4\sum_{a \in \mathcal{A}} \pi_\theta(a|s)\left[-\log \pi_\theta(a|s)\right]^2 + \frac{2(\log A)^2}{(1-\gamma)^2}\left[1 + 2\gamma^2\right] \overset{(c)}{\leq} \frac{16}{e^2}A + \frac{6(\log A)^2}{(1-\gamma)^2}$$

where $(a)$ uses the Bellman equation and equation 50. Inequality $(b)$ follows from the realization that equation 50 is satisfied by every $s \in \mathcal{S}$. Finally, $(c)$ uses the fact that $x^2 \exp(-x) \leq 4e^{-2}, \forall x \geq 0$. Combining equation 49 and equation 51, we arrive at,

$$\mathbf{E}_{a \sim \pi_\theta(s)}\left[\left|A^{\pi_\theta}_{r+\lambda c+\tau\psi_\theta}(s, a)\right|^2\right] \leq 2\mathbf{E}_{a \sim \pi_\theta(s)}\left[\left|A^{\pi_\theta}_{r+\lambda c}(s, a)\right|^2\right] + 2\tau^2 \mathbf{E}_{a \sim \pi_\theta(s)}\left[\left|A^{\pi_\theta}_{\psi_\theta}(s, a)\right|^2\right]$$
$$\leq \frac{8(1+\lambda)^2}{(1-\gamma)^2} + \tau^2\left[\frac{32}{e^2}A + \frac{12(\log A)^2}{(1-\gamma)^2}\right] \tag{52}$$

This concludes the lemma.

## C  Proof of Lemma 4

Fix a tuple $(\theta, \lambda, \tau, \omega)$. The first statement can be proven by observing that,

$$\mathbf{E}\left[\hat{J}_{c,\rho}(\theta)|\theta\right] = (1-\gamma)\mathbf{E}\left[\sum_{t=0}^\infty \gamma^t \sum_{j=0}^t c(s_j, a_j)\Big|s_0 \sim \rho, \pi_\theta\right]$$
$$= (1-\gamma)\mathbf{E}\left[\sum_{j=0}^\infty c(s_j, a_j)\sum_{t=j}^\infty \gamma^t\Big|s_0 \sim \rho, \pi_\theta\right] = \mathbf{E}\left[\sum_{j=0}^\infty \gamma^j c(s_j, a_j)\Big|s_0 \sim \rho, \pi_\theta\right] = J_{c,\rho}(\theta) \tag{53}$$

To prove the second statement, we will first show that the state $\hat{s}$ generated by Algorithm 1 is indeed distributed according to the state-occupancy measure $d^{\pi_\theta}$. Note that,

$$\Pr(\hat{s} = s|\rho, \pi_\theta) = (1-\gamma)\sum_{t=0}^\infty \gamma^t \Pr(s_t = s|s_0 \sim \rho, \pi_\theta) = d^{\pi_\theta}(s) \tag{54}$$

Next, we will show that the $Q$ and $V$ function estimates obtained by Algorithm 1 for arbitrary utility function, $g$, are unbiased. Note the following chain of equalities for a given state-action pair $(s, a)$.

$$\mathbf{E}\left[\hat{Q}_g^{\pi_\theta}(s,a)\big|\theta,s,a\right] = (1-\gamma)\mathbf{E}\left[\sum_{t=0}^\infty \gamma^t \sum_{j=0}^t g(s_i,a_i)\bigg|s_0=s,a_0=a,\pi_\theta\right]$$

$$= (1-\gamma)\mathbf{E}\left[\sum_{j=0}^\infty g(s_i,a_i)\sum_{t=j}^\infty \gamma^t\bigg|s_0=s,a_0=a,\pi_\theta\right] \tag{55}$$

$$= \mathbf{E}\left[\sum_{j=0}^\infty \gamma^i g(s_i,a_i)\bigg|s_0=s,a_0=a,\pi_\theta\right] = Q_g^{\pi_\theta}(s,a)$$

Similarly, we can also demonstrate that, $\mathbf{E}[\hat{V}_g^{\pi_\theta}(s)|\theta,s] = V_g^{\pi_\theta}(s)$. It, hence, follows that $\mathbf{E}[\hat{A}_g^{\pi_\theta}(s,a)|\theta,s,a] = A_g^{\pi_\theta}(s,a)$ for each $(s,a)$. Combining these results, we deduce the following.

$$\mathbf{E}\left[\hat{F}(\theta)\big|\theta\right] = \mathbf{E}_{\hat{s}\sim d^{\pi_\theta}}\mathbf{E}_{a\sim\pi_\theta(\hat{s})}\left[\nabla_\theta\log\pi_\theta(a|\hat{s})\otimes\nabla_\theta\log\pi_\theta(a|\hat{s})\big|\theta\right] = F(\theta), \tag{56}$$

$$\mathbf{E}\left[\hat{H}_\tau(\theta,\lambda)\big|\theta,\lambda\right] = \mathbf{E}_{\hat{s}\sim d^{\pi_\theta}}\mathbf{E}_{a\sim\pi_\theta(\hat{s})}\left[\mathbf{E}\left[\hat{A}_{r+\lambda c+\tau\psi_\theta}^{\pi_\theta}(\hat{s},a)\bigg|\theta,\lambda,\hat{s},a\right]\nabla_\theta\log\pi_\theta(a|\hat{s})\bigg|\theta,\lambda\right]$$

$$= \mathbf{E}_{(\hat{s},a)\sim\nu_\rho^{\pi_\theta}}\left[A_{r+\lambda c+\tau\psi_\theta}^{\pi_\theta}(\hat{s},a)\nabla_\theta\log\pi_\theta(a|\hat{s})\big|\theta,\lambda\right] = H_\tau(\theta,\lambda) \tag{57}$$

The unbiasedness of the gradient estimator can be proved by combining the above two results.

## D  Proof of Lemma 5

Recall the definition of $H_\tau(\theta,\lambda)$ in Lemma 2. Using the Cauchy-Schwarz inequality, we get,

$$\|H_\tau(\theta,\lambda)\|^2 \le \mathbf{E}_{(s,a)\sim\nu^{\pi_\theta}}\left[\left|A_{r+\lambda c+\tau\psi_\theta}^{\pi_\theta}(s,a)\right|^2\|\nabla_\theta\log\pi_\theta(a|s)\|^2\right]$$

$$\overset{(a)}{\le} G^2\mathbf{E}_{s\sim d^{\pi_\theta}}\mathbf{E}_{a\sim\pi_\theta(s)}\left[\left|A_{r+\lambda c+\tau\psi_\theta}^{\pi_\theta}(s,a)\right|^2\right]$$

where $(a)$ follows from Assumption 2. The lemma can now be concluded using Lemma 2 and 3.

## E  Proof of Lemma 6

**Step 1:** We use the following notations: $\omega_{k,\tau}^* \triangleq \omega_\tau^*(\theta_k,\lambda_k)$, $\psi_k(s,a) \triangleq -\log\pi_{\theta_k}(a|s)$, and $\psi_\tau^*(s,a) \triangleq -\log\pi_\tau^*(a|s)$, $\forall(s,a)$. Observe the following chain of inequalities.

$$\mathrm{KL}_k^\tau - \mathrm{KL}_{k+1}^\tau = \mathbf{E}_{(s,a)\sim\nu_\tau^{\pi_*}}\left[\log\frac{\pi_{\theta_{k+1}}(a|s)}{\pi_{\theta_k}(a|s)}\right]$$

$$\overset{(a)}{\ge} \mathbf{E}_{(s,a)\sim\nu_\tau^{\pi_*}}[\nabla_\theta\log\pi_{\theta_k}(a|s)\cdot(\theta_{k+1}-\theta_k)] - \frac{B}{2}\|\theta_{k+1}-\theta_k\|^2$$

$$\overset{(b)}{=} \eta\mathbf{E}_{(s,a)\sim\nu_\tau^{\pi_*}}[\nabla_\theta\log\pi_{\theta_k}(a|s)\cdot\omega_k] - \frac{B\eta^2}{2}\|\omega_k\|^2$$

$$= \eta\mathbf{E}_{(s,a)\sim\nu_\tau^{\pi_*}}\left[\nabla_\theta\log\pi_{\theta_k}(a|s)\cdot\omega_{k,\tau}^*\right] + \eta\mathbf{E}_{(s,a)\sim\nu_\tau^{\pi_*}}[\nabla_\theta\log\pi_{\theta_k}(a|s)\cdot(\omega_k-\omega_{k,\tau}^*)] - \frac{B\eta^2}{2}\|\omega_k\|^2 \tag{58}$$

$$\overset{(b)}{=} \eta\underbrace{\left[J_{r+\lambda_k c+\tau\psi_k}^{\pi_\tau^*} - J_{r+\lambda_k c+\tau\psi_k}^{\pi_{\theta_k}}\right]}_{T_0} + \eta\underbrace{\mathbf{E}_{(s,a)\sim\nu_\tau^{\pi_*}}\left[\nabla_\theta\log\pi_{\theta_k}(a|s)\cdot\omega_{k,\tau}^* - \frac{1}{1-\gamma}A_{r+\lambda_k c+\tau\psi_k}^{\pi_{\theta_k}}\right]}_{T_1}$$

$$+ \eta\mathbf{E}_{(s,a)\sim\nu_\tau^{\pi_*}}[\nabla_\theta\log\pi_{\theta_k}(a|s)\cdot(\omega_k-\omega_{k,\tau}^*)] - \frac{B\eta^2}{2}\|\omega_k\|^2$$

where $(a)$ follows from Assumption 2 and $(b)$, $(c)$ utilize equation 26 and Lemma 9 respectively. Using Assumption 3 and the Cauchy-Schwarz inequality, we derive the following.

$$T_1 \geq - \left\{ \mathbf{E}_{(s,a)\sim\nu^{\pi^*_\tau}} \left[ \nabla_\theta \log \pi_{\theta_k}(a|s) \cdot \omega^*_{k,\tau} - \frac{1}{1-\gamma} A^{\pi_{\theta_k}}_{r+\lambda_k c + \tau \psi_k} \right]^2 \right\}^{\frac{1}{2}} \geq -\sqrt{\epsilon_{\text{bias}}} \tag{59}$$

Moreover, the term $T_0$ can be decomposed as follows,

$$
\begin{aligned}
T_0 &= \left[ J^{\pi^*_\tau}_{r+\lambda_k c + \tau \psi^*_\tau} - J^{\pi_{\theta_k}}_{r+\lambda_k c + \tau \psi_k} \right] + \left[ J^{\pi^*_\tau}_{r+\lambda_k c + \tau \psi_k} - J^{\pi^*_\tau}_{r+\lambda_k c + \tau \psi^*_\tau} \right] \\
&= J^{\pi^*_\tau}_{r+\lambda_k c + \tau \psi^*_\tau} - J^{\pi_{\theta_k}}_{r+\lambda_k c + \tau \psi_k} + \frac{\tau}{1-\gamma} \text{KL}^\tau_k \\
&\geq J^{\pi^*_\tau}_{r+\lambda_k c + \tau \psi^*_\tau} - J^{\pi_{\theta_k}}_{r+\lambda_k c + \tau \psi_k} + \tau \text{KL}^\tau_k
\end{aligned}
\tag{60}
$$

Combining the above results and taking expectations, we obtain,

$$
\begin{aligned}
\mathbf{E}\left[\text{KL}^\tau_{k+1}\right] &\leq (1-\eta\tau)\mathbf{E}\left[\text{KL}^\tau_k\right] + \eta\mathbf{E}\left[ J^{\pi_{\theta_k}}_{r+\lambda_k c + \tau \psi_k} - J^{\pi^*_\tau}_{r+\lambda_k c + \tau \psi^*_\tau} \right] \\
&\quad + \eta\sqrt{\epsilon_{\text{bias}}} + \eta\mathbf{E}_{(s,a)\sim\nu^{\pi^*_\tau}}\mathbf{E}\left[ \nabla_\theta \log \pi_{\theta_k}(a|s) \cdot (\omega^*_{k,\tau} - \mathbf{E}\left[\omega_k | \theta_k, \lambda_k\right]) \right] + \frac{B\eta^2}{2}\mathbf{E}\|\omega_k\|^2 \\
&\stackrel{(a)}{\leq} (1-\eta\tau)\mathbf{E}\left[\text{KL}^\tau_k\right] + \eta\mathbf{E}\left[ J^{\pi_{\theta_k}}_{r+\lambda_k c + \tau \psi_k} - J^{\pi^*_\tau}_{r+\lambda_k c + \tau \psi^*_\tau} \right] \\
&\quad + \eta\sqrt{\epsilon_{\text{bias}}} + \eta G\mathbf{E}\left\|\omega^*_{k,\tau} - \mathbf{E}\left[\omega_k | \theta_k, \lambda_k\right]\right\| + \frac{B\eta^2}{2}\mathbf{E}\|\omega_k\|^2
\end{aligned}
\tag{61}
$$

where the inequality $(a)$ is due to Assumption 2.

**Step 2:** Here, the dual update equation will be utilized. Observe the following.

$$
\begin{aligned}
\eta\mathbf{E}\left[\mathcal{L}_\tau(\theta_k, \lambda_k) - \mathcal{L}_\tau(\theta_k, \lambda^*_\tau)\right] &= \eta\mathbf{E}\left[ (\lambda_k - \lambda^*_\tau)J_c(\theta_k) + \frac{\tau}{2}(\lambda_k)^2 - \frac{\tau}{2}(\lambda^*_\tau)^2 \right] \\
&= \eta\mathbf{E}\left[ (\lambda_k - \lambda^*_\tau)(J_c(\theta_k) + \tau\lambda_k) \right] - \frac{\eta\tau}{2}\mathbf{E}\left[(\lambda_k - \lambda^*_\tau)^2\right]
\end{aligned}
\tag{62}
$$

Using the dual update equation 27 and the non-expansiveness of the projection operator $\mathcal{P}_\Lambda$, we can write the following.

$$
\begin{aligned}
\frac{1}{2}\mathbf{E}\left[(\lambda_{k+1} - \lambda^*_\tau)^2\right] &\leq \frac{1}{2}\mathbf{E}\left[ \left\{ \lambda_k - \lambda^*_\tau - \eta\left(\hat{J}_c(\theta_k) + \tau\lambda_k\right) \right\}^2 \right] \\
&= \frac{1}{2}\mathbf{E}\left[(\lambda_k - \lambda^*_\tau)^2\right] - \eta\mathbf{E}\left[ (\lambda_k - \lambda^*_\tau)\left(\hat{J}_c(\theta_k) + \tau\lambda_k\right) \right] + \frac{\eta^2}{2}\mathbf{E}\left[ \left(\hat{J}_c(\theta_k) + \tau\lambda_k\right)^2 \right] \\
&\stackrel{(a)}{\leq} \frac{1}{2}\mathbf{E}\left[(\lambda_k - \lambda^*_\tau)^2\right] - \eta\mathbf{E}\left[ (\lambda_k - \lambda^*_\tau)(J_c(\theta_k) + \tau\lambda_k) \right] + \eta^2\mathbf{E}\left[\hat{J}^2_c(\theta_k)\right] + \eta^2\tau^2\lambda^2_{\max}
\end{aligned}
\tag{63}
$$

where $(a)$ uses Lemma 4, the fact that $\lambda_k$ and $\hat{J}_c(\theta_k)$ are conditionally independent given $\theta_k$ and $(a+b)^2 \leq 2(a^2+b^2)$ for any $a, b$. Furthermore, Algorithm 1 states that $\hat{J}^2_c(\theta_k)$ can take a value of at most $(j+1)^2$ with probability $(1-\gamma)\gamma^j$, $j \in \{0, 1, \cdots\}$. This leads to the following bound.

$$\mathbf{E}\left[\hat{J}^2_c(\theta_k)\right] \leq \sum_{j=0}^{\infty}(1-\gamma)(j+1)^2\gamma^j = \frac{1+\gamma}{(1-\gamma)^2} < \frac{2}{(1-\gamma)^2} \tag{64}$$

Combining the above results, we finally obtain,

$$
\begin{aligned}
\frac{1}{2}\mathbf{E}\left[(\lambda_{k+1} - \lambda^*_\tau)^2\right] &\leq \frac{1}{2}(1-\eta\tau)\mathbf{E}\left[(\lambda_k - \lambda^*_\tau)^2\right] + \eta\mathbf{E}\left[\mathcal{L}_\tau(\theta_k, \lambda^*_\tau) - \mathcal{L}_\tau(\theta_k, \lambda_k)\right] \\
&\quad + \eta^2\left[ \frac{2}{(1-\gamma)^2} + \tau^2\lambda^2_{\max} \right]
\end{aligned}
\tag{65}
$$

**Step 3:** It is easy to verify that

$$\left[ J_{r+\lambda_k c+\tau\psi_k}^{\pi_{\theta_k}} - J_{r+\lambda_k c+\tau\psi_\tau^*}^{\pi_\tau^*} \right] + \left[ \mathcal{L}_\tau(\theta_k, \lambda_\tau^*) - \mathcal{L}_\tau(\theta_k, \lambda_k) \right] = \mathcal{L}_\tau(\theta_k, \lambda_\tau^*) - \mathcal{L}_\tau(\pi_\tau^*, \lambda_k) \tag{66}$$

Adding equation 61 and equation 65, and using the definition of $\Phi_k^\tau$, we obtain the following inequality.

$$\Phi_{k+1}^\tau \leq (1 - \eta\tau)\Phi_k^\tau + \eta\sqrt{\epsilon_{\text{bias}}} + \eta G \mathbf{E} \left\| \omega_{k,\tau}^* - \mathbf{E}\left[ \omega_k | \theta_k, \lambda_k \right] \right\| + \frac{B\eta^2}{2} \mathbf{E}\|\omega_k\|^2$$
$$+ \eta^2 \left[ \frac{2}{(1-\gamma)^2} + \tau^2\lambda_{\max}^2 \right] + \eta \underbrace{\mathbf{E}\left[ \mathcal{L}_\tau(\theta_k, \lambda_\tau^*) - \mathcal{L}_\tau(\pi_\tau^*, \lambda_k) \right]}_{T_2} \tag{67}$$

The lemma can be proven by showing that $T_2 \leq 0$ which is a direct consequence of Lemma 10. In particular,

$$\mathcal{L}_\tau(\pi_{\theta_k}, \lambda_\tau^*) \leq \max_{\pi \in \Pi} \mathcal{L}_\tau(\pi, \lambda_\tau^*) = \min_{\lambda \geq 0} \mathcal{L}_\tau(\pi_\tau^*, \lambda) \leq \mathcal{L}_\tau(\pi_\tau^*, \lambda_k) \tag{68}$$

## F  Proof of Lemma 7

Define a function $\hat{\zeta}_{\theta,\lambda}^\tau : \mathcal{S} \times \mathcal{A} \to \mathbb{R}$ as follows.

$$\hat{\zeta}_{\theta,\lambda}^\tau(s,a) \triangleq \nabla_\theta \log \pi_\theta(a|s) \cdot \omega_\tau^*(\theta, \lambda) - \frac{1}{1-\gamma} \hat{A}_{r+\lambda c+\tau\psi_\theta}^{\pi_\theta}(s,a), \; \forall (s,a) \tag{69}$$

From equation 13, the gradient estimation can be written as follows.

$$\hat{\nabla}_\omega \mathcal{E}_{\nu^{\pi_\theta}}^\tau(\omega_\tau^*(\theta,\lambda), \theta, \lambda) = \mathbf{E}_{a \sim \pi_\theta(\hat{s})} \left[ \hat{\zeta}_{\theta,\lambda}^\tau(\hat{s}, a) \nabla_\theta \log \pi_\theta(a|\hat{s}) \right] \tag{70}$$

where the distribution of $\hat{s}$ can be shown to be $d^{\pi_\theta}$ (see the proof of Lemma 4). Therefore,

$$\mathbf{E}\left[ \hat{\nabla}_\omega \mathcal{E}_{\nu^{\pi_\theta}}^\tau(\omega_\tau^*(\theta,\lambda), \theta, \lambda) \otimes \hat{\nabla}_\omega \mathcal{E}_{\nu^{\pi_\theta}}^\tau(\omega_\tau^*(\theta,\lambda), \theta, \lambda) \right]$$
$$= \mathbf{E}_{s \sim d^{\pi_\theta}} \hat{\mathbf{E}}_s^\theta \left[ \mathbf{E}_{a \sim \pi_\theta(s)} \left[ \hat{\zeta}_{\theta,\lambda}^\tau(s,a) \nabla_\theta \log \pi_\theta(a|s) \right] \otimes \mathbf{E}_{a \sim \pi_\theta(s)} \left[ \hat{\zeta}_{\theta,\lambda}^\tau(s,a) \nabla_\theta \log \pi_\theta(a|s) \right] \right]$$
$$\preceq \mathbf{E}_{s \sim d^{\pi_\theta}} \left[ \hat{\mathbf{E}}_s^\theta \left[ \mathbf{E}_{a \sim \pi_\theta(s)} \left[ \left( \hat{\zeta}_{\theta,\lambda}^\tau(s,a) \right)^2 \right] \right] \mathbf{E}_{a \sim \pi_\theta(s)} \left[ \nabla_\theta \log \pi_\theta(a|s) \otimes \log \pi_\theta(a|s) \right] \right] \tag{71}$$
$$= \mathbf{E}_{s \sim d^{\pi_\theta}} \left[ \mathbf{E}_{a \sim \pi_\theta(s)} \left[ \hat{\mathbf{E}}_{s,a}^\theta \left[ \left( \hat{\zeta}_{\theta,\lambda}^\tau(s,a) \right)^2 \right] \right] \mathbf{E}_{a \sim \pi_\theta(s)} \left[ \nabla_\theta \log \pi_\theta(a|s) \otimes \log \pi_\theta(a|s) \right] \right]$$

where the semidefinite inequality is a consequence of the Cauchy-Schwarz inequality and the expectation $\hat{\mathbf{E}}_s^\theta$ is computed over the distribution of all sample paths originating from $s$ that are used to estimate $\{\hat{A}_{r+\lambda c+\tau\psi_\theta}^{\pi_\theta}(s,a)\}_{a \in \mathcal{A}}$. In a similar way, $\hat{\mathbf{E}}_{s,a}^\theta$ is defined as the expectation over the distribution of all sample paths that are used in estimating $\hat{A}_{r+\lambda c+\tau\psi_\theta}^{\pi_\theta}(s,a)$. Note that,

$$\left( \hat{\zeta}_{\theta,\lambda}^\tau(s,a) \right)^2 \leq 2 \left\| \nabla_\theta \log \pi_\theta(a|s) \right\|^2 \left\| \omega_\tau^*(\theta, \lambda) \right\|^2 + \frac{2}{(1-\gamma)^2} \left| \hat{A}_{r+\lambda c+\tau\psi_\theta}^{\pi_\theta}(s,a) \right|^2$$
$$\overset{(a)}{\leq} 2G^2 \left\| F(\theta)^\dagger \nabla_\theta \mathcal{L}_\tau(\theta, \lambda) \right\|^2 + \frac{6}{(1-\gamma)^2} \left\{ \left[ \hat{A}_r^{\pi_\theta}(s,a) \right]^2 + \lambda^2 \left[ \hat{A}_c^{\pi_\theta}(s,a) \right]^2 + \tau^2 \left[ \hat{A}_{\psi_\theta}^{\pi_\theta}(s,a) \right]^2 \right\} \tag{72}$$
$$\overset{(b)}{\leq} \frac{2G^4}{\mu_F^2(1-\gamma)^2} L_{\tau,\lambda}^2 + \frac{6}{(1-\gamma)^2} \left\{ \left[ \hat{A}_r^{\pi_\theta}(s,a) \right]^2 + \lambda^2 \left[ \hat{A}_c^{\pi_\theta}(s,a) \right]^2 + \tau^2 \left[ \hat{A}_{\psi_\theta}^{\pi_\theta}(s,a) \right]^2 \right\}$$

where $(a)$ follows from Assumption 2 and the definition of $\omega_\tau^*(\theta, \lambda)$ whereas $(b)$ results from Assumption 4 and Lemma 5. Note that, $\forall g \in \{r, c\}$, and arbitrary $(s,a)$, both $[\hat{Q}_g^{\pi_\theta}(s,a)]^2$ and $[\hat{V}_g^{\pi_\theta}(s)]^2$ can assume a

value of at most $(j+1)^2$ with probability $(1-\gamma)\gamma^j$, $j \in \{0, 1, \cdots\}$ (see Algorithm 1). Therefore, the following holds $\forall(s, a)$.

$$
\begin{aligned}
\max_{g \in \{r,c\}} \left\{ \hat{\mathbf{E}}_{s,a}^\theta \left[ \left( \hat{A}_g^{\pi_\theta}(s,a) \right)^2 \right] \right\} &\leq 2 \max_{g \in \{r,c\}} \left\{ \hat{\mathbf{E}}_{s,a}^\theta \left[ \left( \hat{Q}_g^{\pi_\theta}(s,a) \right)^2 \right] + \hat{\mathbf{E}}_{s,a}^\theta \left[ \left( \hat{V}_g^{\pi_\theta}(s) \right)^2 \right] \right\} \\
&\leq 4 \max_{g \in \{r,c\}} \left\{ \max \left\{ \hat{\mathbf{E}}_{s,a}^\theta \left[ \left( \hat{Q}_g^{\pi_\theta}(s,a) \right)^2 \right], \hat{\mathbf{E}}_{s,a}^\theta \left[ \left( \hat{V}_g^{\pi_\theta}(s) \right)^2 \right] \right\} \right\} \\
&\leq 4(1-\gamma) \sum_{j=0}^\infty (j+1)^2 \gamma^j = \frac{4(1+\gamma)}{(1-\gamma)^2} < \frac{8}{(1-\gamma)^2}
\end{aligned}
\tag{73}
$$

Moreover, observe that the following inequalities hold for arbitrary $s \in \mathcal{S}$.

$$
\begin{aligned}
\mathbf{E}_{a \sim \pi_\theta(s)} \hat{\mathbf{E}}_{s,a}^\theta \left[ \left( \hat{Q}_{\psi_\theta}^{\pi_\theta}(s,a) \right)^2 \right] &= (1-\gamma) \sum_{j=0}^\infty \gamma^j \sum_{a \in \mathcal{A}} \pi_\theta(a|s) \mathbf{E}_{j,s,a}^\theta \left[ \left\{ \sum_{t=0}^j -\log \pi_\theta(a_t|s_t) \right\}^2 \right] \\
&\overset{(a)}{\leq} (1-\gamma) \sum_{j=0}^\infty (j+1)\gamma^j \sum_{a \in \mathcal{A}} \pi_\theta(a|s) \mathbf{E}_{j,s,a}^\theta \underbrace{\left[ \sum_{t=0}^j (-\log \pi_\theta(a_t|s_t))^2 \right]}_{T}
\end{aligned}
\tag{74}
$$

where $\mathbf{E}_{j,s,a}^\theta$ denotes the expectation over all $j+1$ length $\pi_\theta$-induced trajectories $\{(s_t, a_t)\}_{t=0}^j$ that start with the initial condition $s_0 = s, a_0 = a$. The inequality is a consequence of Cauchy-Schwarz inequality. The term $T$ can be bounded as follows.

$$
\begin{aligned}
T &= \sum_{a \in \mathcal{A}} \pi_\theta(a|s) \left[ \left( -\log \pi_\theta(a|s) \right)^2 + \sum_{t=1}^j \sum_{s' \in \mathcal{S}} \Pr_t^{\pi_\theta}(s'|s,a) \sum_{a' \in \mathcal{A}} \pi_\theta(a'|s') \left( -\log \pi_\theta(a'|s') \right)^2 \right] \\
&\overset{(a)}{\leq} \frac{4}{e^2}(j+1)A
\end{aligned}
\tag{75}
$$

where $\Pr_t^{\pi_\theta}(s'|s,a) = \Pr(s_t = s'|s_0 = s, a_0 = a, \pi_\theta)$. Inequality $(a)$ follows from the fact that $x^2 \exp(-x) \leq 4e^{-2}$, $\forall x \geq 0$. Applying the above bound in equation 74, we deduce,

$$
\mathbf{E}_{a \sim \pi_\theta(s)} \hat{\mathbf{E}}_{s,a}^\theta \left[ \left( \hat{Q}_{\psi_\theta}^{\pi_\theta}(s,a) \right)^2 \right] \leq \frac{4}{e^2} A(1-\gamma) \sum_{j=0}^\infty (j+1)^2 \gamma^j < \frac{8Ae^{-2}}{(1-\gamma)^2}
\tag{76}
$$

In a similar manner, we can also show that,

$$
\mathbf{E}_{a \sim \pi_\theta(s)} \hat{\mathbf{E}}_{s,a}^\theta \left[ \left( \hat{V}_{\psi_\theta}^{\pi_\theta}(s) \right)^2 \right] \leq \frac{8Ae^{-2}}{(1-\gamma)^2}
\tag{77}
$$

Combining the above results, one arrives at,

$$
\begin{aligned}
\mathbf{E}_{a \sim \pi_\theta(s)} \hat{\mathbf{E}}_{s,a}^\theta \left[ \left( \hat{A}_{\psi_\theta}^{\pi_\theta}(s,a) \right)^2 \right] &\leq 2\mathbf{E}_{a \sim \pi_\theta(s)} \hat{\mathbf{E}}_{s,a}^\theta \left[ \left( \hat{Q}_{\psi_\theta}^{\pi_\theta}(s,a) \right)^2 \right] + 2\mathbf{E}_{a \sim \pi_\theta(s)} \hat{\mathbf{E}}_{s,a}^\theta \left[ \left( \hat{V}_{\psi_\theta}^{\pi_\theta}(s) \right)^2 \right] \\
&\leq \frac{32Ae^{-2}}{(1-\gamma)^2}
\end{aligned}
\tag{78}
$$

Combining equation 72, equation 73, and equation 78, we can write,

$$
\mathbf{E}_{a \sim \pi_\theta(s)} \left[ \hat{\mathbf{E}}_{s,a}^\theta \left[ \left( \hat{\zeta}_{\theta,\lambda}^\tau(s,a) \right)^2 \right] \right] \leq \frac{2G^4}{\mu_F^2(1-\gamma)^2} L_{\tau,\lambda}^2 + \frac{48}{(1-\gamma)^4} \left[ 1 + \lambda^2 + 4Ae^{-2}\tau^2 \right] = \sigma_{\tau,\lambda}^2
\tag{79}
$$

Substituting the above bound into equation 71, we conclude the lemma.

# G  Proof of Lemma 8

We prove Lemma 8 using Corollary 2 of Jain et al. (2018). Observe the following three statements for a given $(\lambda, \tau)$.

**S1** : The following quantities exist and are finite $\forall \theta \in \mathbb{R}^d$.

$$F(\theta) \triangleq \mathbf{E}_{(s,a) \sim \nu^{\pi_\theta}} \left[ \nabla_\theta \log \pi_\theta(a|s) \otimes \nabla_\theta \log \pi_\theta(a|s) \right], \tag{80}$$

$$G(\theta) \triangleq \mathbf{E}_{(s,a) \sim \nu^{\pi_\theta}} \left[ \nabla_\theta \log \pi_\theta(a|s) \otimes \nabla_\theta \log \pi_\theta(a|s) \otimes \nabla_\theta \log \pi_\theta(a|s) \otimes \nabla_\theta \log \pi_\theta(a|s) \right] \tag{81}$$

**S2** : There exists $\sigma_{\tau,\lambda}^2$ such that the following is obeyed $\forall \theta \in \mathbb{R}^d$ where $\omega_\tau^*(\theta, \lambda)$ is the minimizer of $\mathcal{E}_{\nu^{\pi_\theta}}^\tau(\cdot, \theta, \lambda)$.

$$\mathbf{E} \left[ \hat{\nabla}_\omega \mathcal{E}_{\nu^{\pi_\theta}}^\tau(\omega_\tau^*(\theta, \lambda), \theta, \lambda) \otimes \hat{\nabla}_\omega \mathcal{E}_{\nu^{\pi_\theta}}^\tau(\omega_\tau^*(\theta, \lambda), \theta, \lambda) \right] \preccurlyeq \sigma_{\tau,\lambda}^2 F(\theta) \tag{82}$$

**S3** : There exists $\mu_F, G > 0$ such that the following statements hold $\forall \theta \in \mathbb{R}^d$.

$$(a) \ F(\theta) \succcurlyeq \mu_F I_d, \tag{83}$$

$$(b) \ \mathbf{E}_{(s,a) \sim \nu^{\pi_\theta}} \left[ \|\nabla_\theta \log \pi_\theta(a|s)\|^2 \nabla_\theta \log \pi_\theta(a|s) \otimes \nabla_\theta \log \pi_\theta(a|s) \right] \preccurlyeq G^2 F(\theta), \tag{84}$$

$$(c) \ \mathbf{E}_{(s,a) \sim \nu^{\pi_\theta}} \left[ \|\nabla_\theta \log \pi_\theta(a|s)\|_{F(\theta)^\dagger}^2 \nabla_\theta \log \pi_\theta(a|s) \otimes \nabla_\theta \log \pi_\theta(a|s) \right] \preccurlyeq \frac{G^2}{\mu_F} F(\theta) \tag{85}$$

Statement **S1** is a consequence of Assumption 2 whereas **S2** results from Lemma 7. Statement **S3**$(a)$ is identical to Assumption 4, **S3**$(b)$ follows from Assumption 2, and finally, **S3**$(c)$ can be deduced via Assumption 2 and 4. We can, hence, apply Corollary 2 of Jain et al. (2018) with $\kappa = \tilde{\kappa} = G^2/\mu_F$ and derive the following result if $H > \bar{C}\sqrt{\kappa\tilde{\kappa}} \log(\sqrt{d}\sqrt{\kappa\tilde{\kappa}})$, the learning rates are set as $\alpha = \frac{3\sqrt{5}\sqrt{\kappa\tilde{\kappa}}}{1 + 3\sqrt{5\kappa\tilde{\kappa}}}$, $\beta = \frac{1}{9\sqrt{\kappa\tilde{\kappa}}}$, $\xi = \frac{1}{3\sqrt{5}\mu_F\sqrt{\kappa\tilde{\kappa}}}$, and $\delta = \frac{1}{5G^2}$, and $\lambda_k \in \Lambda$.

$$\mathbf{E}\left[l_k^\tau(\omega_k)\right] - l_k^\tau(\omega_{k,\tau}^*) \leq \frac{C}{2} \exp\left(-\frac{H}{20\sqrt{\kappa\tilde{\kappa}}}\right) \left[l_k^\tau(\mathbf{0}) - l_k^\tau(\omega_{k,\tau}^*)\right] + 11 \frac{\sigma_{\tau,\lambda_{\max}}^2 d}{H},$$
$$\text{where } l_k^\tau(\omega) \triangleq \mathcal{E}_{\nu^{\pi_{\theta_k}}}^\tau(\omega, \theta_k, \lambda_k), \ \forall \omega \in \mathbb{R}^d, \text{ and } \omega_{k,\tau}^* \triangleq \omega_\tau^*(\theta_k, \lambda_k) \tag{86}$$

In the above inequality, the expectation is over the distribution of $\omega_k$ for a given $(\theta_k, \lambda_k)$. The term $C$ is a universal constant. Note that $l_k^\tau(\omega_{k,\tau}^*) \geq 0$ and $l_k^\tau(\mathbf{0})$ is bounded above as follows.

$$l_k^\tau(\mathbf{0}) = \frac{1}{2} \mathbf{E}_{(s,a) \sim \nu^{\pi_{\theta_k}}} \left[ \frac{1}{1-\gamma} A_{r+\lambda_k c + \tau\psi_k}^{\pi_{\theta_k}}(s,a) \right]^2 \overset{(a)}{\leq} \frac{L_{\tau,\lambda_{\max}}^2}{2(1-\gamma)^2} \tag{87}$$

where $(a)$ follows from Lemma 3. Combining *equation* 86, *equation* 87, and the fact that $l_k(\cdot)$ is $\mu_F$-strongly convex, we establish,

$$\mathbf{E}\|\omega_k - \omega_k^*\|^2 \leq \frac{2}{\mu_F} \left[\mathbf{E}\left[l_k(\omega_k)\right] - l_k(\omega_k^*)\right] \leq 22 \frac{\sigma_{\tau,\lambda_{\max}}^2 d}{\mu_F H} + C \exp\left(-\frac{\mu_F}{20G^2} H\right) \left[\frac{L_{\tau,\lambda_{\max}}^2}{\mu_F(1-\gamma)^2}\right] \tag{88}$$

The first statement can be proven by applying $H \geq 1$ and $\exp(-x) \leq 1, \ \forall x \geq 0$. We get the following for noiseless ($\sigma_{\tau,\lambda_{\max}}^2 = 0$) gradient updates.

$$\mathbf{E}\|(\mathbf{E}[\omega_k|\theta_k] - \omega_k^*)\|^2 \leq C \exp\left(-\frac{\mu_F}{20G^2} H\right) \left[\frac{L_{\tau,\lambda_{\max}}^2}{\mu_F(1-\gamma)^2}\right] \tag{89}$$

The second statement can be established from *equation* 89 by applying Jensen's inequality on the function $f(x) = x^2$.

## H   Proof of Corollary 1

Lemma 6 establishes the following result $\forall k \in \{0, 1, \cdots, K-1\}$.

$$\Phi_{k+1}^{\tau} \leq (1 - \eta\tau)\Phi_k^{\tau} + \eta\sqrt{\epsilon_{\text{bias}}}$$
$$+ \underbrace{\eta G \mathbf{E} \left\| \mathbf{E}\left[\omega_k | \theta_k, \lambda_k\right] - \omega_{k,\tau}^* \right\| + \frac{B\eta^2}{2} \mathbf{E}\|\omega_k\|^2 + \eta^2 \left[\frac{2}{(1-\gamma)^2} + \tau^2 \lambda_{\max}^2\right]}_{T_k} \tag{90}$$

Using equation 32, and Lemma 8, we can bound the term $T_k$ as follows.

$$T_k \leq \eta \exp\left(-\frac{\mu_F}{40G^2}H\right) \underbrace{\left[\frac{G\sqrt{C}L_{\tau,\lambda_{\max}}}{\sqrt{\mu_F}(1-\gamma)}\right]}_{\leq R_0}$$
$$+ \eta^2 \left\{ \underbrace{\frac{B}{\mu_F}\left[22\sigma_{\tau,\lambda_{\max}}^2 d + \frac{\left(C + \mu_F^{-1}G^2\right)L_{\tau,\lambda_{\max}}^2}{(1-\gamma)^2}\right]}_{\leq R_2} + \underbrace{\left[\frac{2}{(1-\gamma)^2} + \tau^2\lambda_{\max}^2\right]}_{\leq R_1} \right\} \tag{91}$$

where the upper bounds $\{R_l\}_{l\in\{0,1,2\}}$ follow from the fact that $\tau \in [0,1]$ and $L_{\tau,\lambda_{\max}}$, $\sigma_{\tau,\lambda_{\max}}$ are strictly increasing functions of $\tau$. Therefore, we obtain the following simplified recursion $\forall k \in \{1, \cdots, K\}$.

$$\Phi_k^{\tau} \leq (1 - \eta\tau)\Phi_{k-1}^{\tau} + \eta\sqrt{\epsilon_{\text{bias}}} + \eta R_0 \exp\left(-\frac{\mu_F}{40G^2}H\right) + \eta^2(R_1 + R_2)$$
$$\overset{(a)}{\leq} (1-\eta\tau)^k \Phi_0^{\tau} + \left[\sum_{l=0}^{k-1}(1-\eta\tau)^l\right] \times \left[\eta\sqrt{\epsilon_{\text{bias}}} + \eta R_0 \exp\left(-\frac{\mu_F}{40G^2}H\right) + \eta^2(R_1 + R_2)\right] \tag{92}$$
$$\overset{(b)}{\leq} \exp(-\eta\tau k)\Phi_0^{\tau} + \frac{1}{\tau}\sqrt{\epsilon_{\text{bias}}} + \frac{1}{\tau}R_0\exp\left(-\frac{\mu_F}{40G^2}H\right) + \frac{\eta}{\tau}(R_1 + R_2)$$

where $(a)$ is obtained by repeatedly applying the recursion and $(b)$ uses the fact that $\sum_{l=0}^{\infty}(1-\eta\tau)^l = 1/\eta\tau$ and $(1-\eta\tau)^k \leq \exp(-\eta\tau k)$ whenever $\eta\tau < 1$. This establishes the corollary.

## I   Proof of Theorem 1

Observe the following chain of inequalities for a given $k \in \{0, \cdots, K\}$ and $g \in \{r, c\}$.

$$J_g^{\pi_\tau^*} - J_g^{\pi_{\theta_k}} \overset{(a)}{=} \frac{1}{1-\gamma}\sum_{s\in\mathcal{S}} d^{\pi_\tau^*}(s)\sum_{a\in\mathcal{A}}\left\{\pi_\tau^*(a|s) - \pi_{\theta_k}(a|s)\right\}Q_g^{\pi_{\theta_k}}(s,a)$$
$$\leq \frac{1}{(1-\gamma)^2}\sum_{s\in\mathcal{S}} d^{\pi_\tau^*}(s)\|\pi_\tau^*(\cdot|s) - \pi_{\theta_k}(\cdot|s)\|_1$$
$$\overset{(b)}{\leq} \frac{1}{(1-\gamma)^2}\sum_{s\in\mathcal{S}} d^{\pi_\tau^*}(s)\sqrt{2\text{KL}\left(\pi_\tau^*(\cdot|s)||\pi_{\theta_k}(\cdot|s)\right)} \tag{93}$$
$$\overset{(c)}{\leq} \frac{1}{(1-\gamma)^2}\sqrt{2\sum_{s\in\mathcal{S}} d^{\pi_\tau^*}(s)\text{KL}\left(\pi_\tau^*(\cdot|s)||\pi_{\theta_k}(\cdot|s)\right)} = \frac{1}{(1-\gamma)^2}\sqrt{2\text{KL}_k^{\tau}}$$

where $(a)$, $(b)$, and $(c)$ are consequences of Lemma 9, the Pinkster's inequality, and the Cauchy-Schwarz inequality respectively. Taking expectation on both sides of the above inequality, we obtain,

$$
\begin{aligned}
J_g^{\pi_\tau^*} - \mathbf{E}\left[J_g^{\pi_{\theta_k}}\right] &\leq \frac{1}{(1-\gamma)^2}\mathbf{E}\left[\sqrt{2\mathrm{KL}_k^\tau}\right] \\
&\stackrel{(a)}{\leq} \frac{1}{(1-\gamma)^2}\sqrt{2\mathbf{E}\left[\mathrm{KL}_k^\tau\right]} \leq \frac{1}{(1-\gamma)^2}\sqrt{2\Phi_k^\tau} \\
&\stackrel{(b)}{\leq} \frac{\sqrt{2}}{(1-\gamma)^2}\left[\exp\left(-\frac{\eta\tau k}{2}\right)\sqrt{\Phi_0^\tau} + \frac{1}{\sqrt{\tau}}(\epsilon_{\mathrm{bias}})^{\frac{1}{4}} + \sqrt{\frac{R_0}{\tau}}\exp\left(-\frac{\mu_F}{80G^2}H\right) + \sqrt{\frac{\eta}{\tau}}\left(\sqrt{R_1} + \sqrt{R_2}\right)\right]
\end{aligned}
\tag{94}
$$

where $(a)$ is a result of Jensen's inequality applied to $f(x) = \sqrt{x}$, $x \geq 0$ while $(b)$ follows from Corollary 1, and the fact that for the choice of $\eta, \tau$ mentioned in the theorem statement, $\tau \in [0,1]$ and $\eta\tau < 1$ for sufficiently small $\epsilon$. Lemma 10 implies

$$
J_r^{\pi^*} - J_r^{\pi_\tau^*} \leq \tau\mathcal{H}(\pi_\tau^*) \leq \frac{\tau}{(1-\gamma)}\log A
\tag{95}
$$

Combining equation 94 and equation 95, one can obtain an upper bound on the optimality gap. To derive a similar bound for the constraint violation, observe that Lemma 10 implies that,

$$
-(\lambda_{\max} - \lambda_\tau^*)J_c^{\pi_\tau^*} \leq \frac{\tau}{2}\lambda_{\max}^2
\tag{96}
$$

Additionally, since $\lambda_\tau^* = \arg\min_{\lambda \in \Lambda}\mathcal{L}_\tau(\pi_\tau^*, \lambda)$, we have,

$$
\lambda_\tau^* = \arg\min_{\lambda \in \Lambda}\left\{\lambda J_c^{\pi_\tau^*} + \frac{\tau}{2}\lambda^2\right\}
\tag{97}
$$

Therefore, $\lambda_\tau^*$ can assume three possible values.

**Case 1:** If $0 < -\frac{J_c^{\pi_\tau^*}}{\tau} < \frac{\lambda_{\max}}{2}$, we have $\lambda_\tau^* = -\frac{J_c^{\pi_\tau^*}}{\tau}$, and consequently $\lambda_{\max} - \lambda_\tau^* > \frac{\lambda_{\max}}{2}$.

**Case 2:** If $-\frac{J_c^{\pi_\tau^*}}{\tau} \leq 0$, we get $\lambda_\tau^* = 0$ which ensures $\lambda_{\max} - \lambda_\tau^* = \lambda_{\max} > \frac{\lambda_{\max}}{2}$.

For cases 1 and 2, the following result can be derived using equation 96.

$$
-J_c^{\pi_\tau^*} \leq \tau\lambda_{\max}
\tag{98}
$$

**Case 3:** If $\frac{\lambda_{\max}}{2} \leq -\frac{J_c^{\pi_\tau^*}}{\tau}$, either $\lambda_\tau^* = -\frac{J_c^{\pi_\tau^*}}{\tau}$ or $\lambda_\tau^* = \lambda_{\max}$. In both cases, $\lambda_\tau^* \geq \frac{\lambda_{\max}}{2}$. Applying Lemma 10, we get,

$$
\lambda_\tau^*(J_c^{\pi^*} - J_c^{\pi_\tau^*}) \leq J_r^{\pi_\tau^*} - J_r^{\pi^*} + \tau\mathcal{H}(\pi_\tau^*)
\tag{99}
$$

Moreover, using Lemma 1, we have,

$$
J_r^{\pi_\tau^*} - J_r^{\pi^*} \leq \lambda^*(J_c^{\pi^*} - J_c^{\pi_\tau^*})
\tag{100}
$$

Combining equation 99 and equation 100, we arrive at,

$$
(\lambda_\tau^* - \lambda^*)(J_c^{\pi^*} - J_c^{\pi_\tau^*}) \leq \tau\mathcal{H}(\pi_\tau^*)
\tag{101}
$$

Note that $\lambda_\tau^* \geq \frac{\lambda_{\max}}{2}$, and Lemma 1 dictates that $\lambda^* \leq \frac{\lambda_{\max}}{4}$ for the specific choice of $\lambda_{\max}$ given in the theorem statement. Clearly, $\lambda_\tau^* - \lambda^* \geq \frac{\lambda_{\max}}{4}$. Also, $J_c^{\pi^*} \geq 0$ due to feasibility of the optimal solution. Hence,

$$
-J_c^{\pi_\tau^*} \leq \frac{4\tau\mathcal{H}(\pi_\tau^*)}{\lambda_{\max}} \leq \frac{\tau}{(1-\gamma)}\log A
\tag{102}
$$

where the last inequality follows from the fact that $4/\lambda_{\max} = (1-\gamma)c_{\text{slat}} \leq 1$. Combining equation 98 and equation 102, we obtain,

$$-J_c^{\pi_\tau^*} \leq \frac{\tau}{(1-\gamma)} \max\left\{\log A, 4c_{\text{slat}}^{-1}\right\} \tag{103}$$

### I.1 Choice of Parameters

If we take $\tau = \max\{(\epsilon_{\text{bias}})^{\frac{1}{6}}, \epsilon\}$, the following bounds result from equation 95 and equation 103.

$$J_r^{\pi^*} - J_r^{\pi_\tau^*} = \mathcal{O}((\epsilon_{\text{bias}})^{\frac{1}{6}} + \epsilon), \text{ and } -J_c^{\pi_\tau^*} = \mathcal{O}((\epsilon_{\text{bias}})^{\frac{1}{6}} + \epsilon) \tag{104}$$

Moreover, if $\eta = \epsilon^2\tau$, $H = 40G^2\mu_F^{-1}\log(\tau^{-1}\epsilon^{-2})$, and $K = 2\tau^{-2}\epsilon^{-2}$, then equation 94 reduces to the following $\forall g \in \{r, c\}$.

$$J_g^{\pi_\tau^*} - \mathbf{E}[J_g^{\pi_{\theta_K}}] = \mathcal{O}(\epsilon + (\epsilon_{\text{bias}})^{\frac{1}{6}}) \tag{105}$$

To prove the above relation, we utilized the facts that (a) if $\epsilon < (\epsilon_{\text{bias}})^{\frac{1}{6}}$, we have $\tau = (\epsilon_{\text{bias}})^{\frac{1}{6}}$ and $(\epsilon_{\text{bias}})^{\frac{1}{4}}/\sqrt{\tau} = (\epsilon_{\text{bias}})^{\frac{1}{6}}$, and (b) if $\epsilon \geq (\epsilon_{\text{bias}})^{\frac{1}{6}}$, we have $\tau = \epsilon$, leading to $(\epsilon_{\text{bias}})^{\frac{1}{4}}/\sqrt{\tau} \leq \epsilon$. Note that in both of these cases, $(\epsilon_{\text{bias}})^{\frac{1}{4}}/\sqrt{\tau} \leq (\epsilon_{\text{bias}})^{\frac{1}{6}} + \epsilon$. Conjoining equation 104 and equation 105, we conclude the theorem. The criterion that $H$ is sufficiently large is automatically satisfied if $\epsilon$ is sufficiently small.

## J Auxiliary Lemmas

**Lemma 9.** *(Agarwal et al., 2021, Lemma 2) For any two policies $\pi_1, \pi_2$ and a utility $g \in \mathbb{R}^{\mathcal{S} \times \mathcal{A}}$, the following holds.*

$$J_g^{\pi_1} - J_g^{\pi_2} = \frac{1}{1-\gamma}\mathbf{E}_{(s,a)\sim\nu^{\pi_1}}\left[A_g^{\pi_2}(s,a)\right] = \frac{1}{1-\gamma}\sum_{s\in\mathcal{S}} d^{\pi_1}(s)\sum_{a\in\mathcal{A}}\left\{\pi_1(a|s) - \pi_2(a|s)\right\}Q_g^{\pi_2}(s,a) \tag{106}$$

**Lemma 10.** *(Ding et al., 2024, Lemma 14) The primal-dual pair $(\pi_\tau^*, \lambda_\tau^*)$ defined in equation 9 always exists and is unique for $\tau > 0$. Moreover, it satisfies the following property.*

$$\max_{\pi\in\Pi}\mathcal{L}_\tau(\pi, \lambda_\tau^*) = \mathcal{L}_\tau(\pi_\tau^*, \lambda_\tau^*) = \min_{\lambda\in\Lambda}\mathcal{L}_\tau(\pi_\tau^*, \lambda) \tag{107}$$

*Consequently, the following inequalities hold $\forall\pi \in \Pi$ and $\forall\lambda \in \Lambda$.*

$$J_{r+\lambda_\tau^*c}^\pi - \tau\mathcal{H}(\pi_\tau^*) \leq J_{r+\lambda_\tau^*c}^{\pi_\tau^*} \leq J_{r+\lambda c}^{\pi_\tau^*} + \frac{\tau}{2}\lambda^2 \tag{108}$$

