# OpenReview forum: "Last-Iterate Convergence of General Parameterized Policies in Constrained MDPs"
_TMLR — Accepted by TMLR_

### Review · Reviewer_4KU4 · 2026-02-23

**Summary Of Contributions:**

This paper propose a method for solving constrained discounted Markov decision processes with general parameterized policies. The authors propose a primal-dual based regularized accelerated natural policy gradient method that achieves $\tilde{\mathcal{O}}(\epsilon^{-2} \min{\epsilon^{-2}, \epsilon_{\text{bias}}^{-1/3}})$ error for the general case, which is significantly better than prior works in terms of sample complexity in the case of general policy classes, and extends the policy parameterization class if the complexity is similar (Gladden et al., 2023) or worse (Ying et al., 2022). The author applies an interesting technique on the algorithm part, in particular, showing in Lemma 4, that the estimates are unbiased and the argument following eq 38, which are insightful arguments. The algorithm design itself follows primal dual regularized (accelerated) first-order method, in which some results are based on prior work of Jain et al, 2018.

The paper is well written with strong argument and sufficient details. I checked majority of the proof but may miss some details, but as I as I can tell the proof looks solid and the results are reasonable. In particular, lemma 2 is standard. lemma 3 is interesting on its own. Lemma 4 is nice and interesting. Lemma 5 is standard. The following lemma and the main results are solid, in particular, the discussion on the connection to algo 1 is interesting and observant.

Overall I find the paper solid and strong. Some weaknesses are as follows, which are somewhat minor:
1. I find the claiming the assumptions common and thus brushing away the justification is a bit lazy. Can the authors say more on each assumption, and especially its boundary, i.e., when does the algorithm fail to satisfy the assumption? In particular, assump 4 is the most elusive one and I hope the authors can say more about it.
2. The results of $\mathcal{O}(\epsilon^{-2})$ is misleading, in the sense that we do not know the approximation error and cannot set the desired accuracy whatsoever.
3. Although common, I find parameterizing finite action space with general function approximations a bit unnecessary. Even in the case of large action space, this algorithm would necessarily need to parameterize the advantage function as well, thus nontrivial development is also needed in that front. In this paper, the authors seem to treat the advantage in the tabular form?

**Audience:**

Yes

**Audience Explanation:**

Constrained MDP is an interesting class of problems. Researcher in control, RL, and robot learning will be interested in its methodology.

**Claims And Evidence:**

Yes

**Claims Explanation:**

This is a theory paper and the claims are supported by the proofs.

**Requested Changes:**

I don't think major changes are needed for this paper, but hope the authors tighten the notation usage, and definition of the operators, such as various notations appearing in section 3. It is common in theory paper to put all the definitions and notations as a separate subsection right at the front of section 2, in which I hope the authors do the same, along side the MDP definition. The notation in algorithm 2 from ASGD is perplexing, and I hope the authors can make it more clear. The arguments following eq 37 is not immediate to me, I encourage the authors to expand and formalize.

---

> ### Author Response · Authors · 2026-03-31
> **Response to Reviewer 4KU4: Part 1**
>
> **Assumption 2:** Assumption 2 essentially dictates that neither the policy function nor its gradient should rapidly change as a result of a slight nudge in $\theta$. In the absence of such a restriction, the policy gradient algorithm cannot converge to a stationary point. We have added this discussion to the manuscript.
>
> **Assumption 3:** Assumption 3 introduces the term $\epsilon_{\mathrm{bias}}$, which can be thought of as the expressivity power of the parameterized policy class. For rich policy classes (such as neural policies with sufficiently large width), $\epsilon_{\mathrm{bias}}$ is insignificantly small. The effect of $\epsilon_{\mathrm{bias}}$ can be seen in the optimality gap and constraint violation, which are $\mathcal{O}(\epsilon + (\epsilon_{\mathrm{bias}})^{\frac{1}{6}})$, as stated in Theorem 1. Therefore, for inadequate policies, $\epsilon_{\mathrm{bias}}$ will be large, leading to poor optimality gap and constraint-violation guarantees.
>
> **Assumption 4:** Assumption 4 concerns the Fisher matrix $F(\theta) = \mathbf{E}\_{(s, a)\sim \nu\^{\pi\_\theta}}[\nabla\_{\theta}\log \pi\_{\theta}(a|s)\otimes \nabla\_{\theta}\log \pi\_{\theta}(a|s)]$. The very definition of $F(\theta)$ ensures that $F(\theta)$ is positive semi-definite, i.e., all its eigenvalues are non-negative. Assumption 4 adds the mild restriction that, not only the eigenvalues of $F(\theta)$ have to be non-negative, they must be strictly larger than a positive quantity $\mu$. This assumption, popularly known as the Fisher non-degeneracy, is central to policy gradient algorithms, as evident by its widespread usage in the literature. Many policy classes, such as Gaussian and exponential with parameterized mean, Cauchy, etc., satisfy this assumption [1]. It is worth noting that softmax policies may not obey this assumption if they are close to being deterministic. However, if the softmax parameters are bounded in a finite interval, then they can be shown to satisfy this assumption (Remark 1 in [2]). A detailed discussion on this topic has been added in the revised manuscript.
>
> **Knowledge of the approximation error:** As mentioned in our paper, some of the hyperparameters of our algorithm indeed depend on the approximation error $\epsilon\_{\mathrm{bias}}$. We also agree that finding the exact value of $\epsilon\_{\mathrm{bias}}$ might be difficult in practice.  However, it might be possible to obtain an upper bound, $\bar{\epsilon}_{\mathrm{bias}}$, of the desired quantity. For example, [3] shows that for a two-layer neural parameterization of width $m$, the function approximation error is $\mathcal{O}(m\^{-1/8})$ (see Theorem A.4 and the subsequent discussion). Our algorithm still works if this bound is used as a proxy of $\epsilon\_{\mathrm{bias}}$. In this scenario, the convergence error and the sampling complexity will be functions of $\bar{\epsilon}\_{\mathrm{bias}}$, instead of $\epsilon\_{\mathrm{bias}}$. In the absence of appropriate upper bounds, we can utilize hyperparameter tuning, a common method for setting unknown hyperparameters for machine learning algorithms. Therefore, we would like to respectfully disagree with the reviewer that our $\mathcal{O}(\epsilon\^{-2})$ result is misleading.
>
> **Function Approximation:** For finite state space $\mathcal{S}$ and finite action space $\mathcal{A}$, one can work with tabular policies, where each policy function $\pi$ can be represented as a table of size $|\mathcal{S}|\times |\mathcal{A}|$. However, if the state space $\mathcal{S}$ is infinite (which is the case in our framework), such representations no longer work. This necessitates policy parameterization, where a parameterized policy $\pi_{\theta}$ operates as a function that, for each input state $s$, generates a distribution $\pi_{\theta}(\cdot|s)$ over $\mathcal{A}$. The advantage function corresponding to the policy $\pi_{\theta}$ and utility $g$ is a function of the form $A_g^{\pi_{\theta}}:\mathcal{S}\times \mathcal{A}\rightarrow \mathbb{R}$. Since we have chosen $\mathcal{S}$ to be infinite, $A_g^{\pi_{\theta}}$ cannot be expressed in a tabular form. Fortunately, we do not need to obtain $A\_g\^{\pi\_{\theta}}(s, a)$ for every possible pair $(s, a)\in \mathcal{S}\times \mathcal{A}$. As stated in Algorithm 1, we only need to estimate the values $\\{A_g\^{\pi\_{\theta}}(\hat{s}, a)\\}\_{a\in \mathcal{A}}$, where $\hat{s}$ is a sample chosen from the occupancy measure $d\^{\pi\_{\theta}}$. We hope this clarifies the confusion regarding the parameterization of the advantage function.
>
> **Notations:** For ease of reference, we have included a table listing the major notations used in our paper. We hope that it will enhance the reading experience.

---

> ### Author Response · Authors · 2026-03-31
> **Response to Reviewer 4KU4: Part 2**
>
> **Discussion following Eq. (37):** We have elaborated the discussion following Eq. (37) in the revised manuscript to better convey the main idea. We have paraphrased the added section below for ease of reference.
>
> In simple terms, due to our unconventional sampling procedure, the LHS of (33) takes the form of $\mathbf{E}[XY]\otimes \mathbf{E}[XY]$, where $X, Y$ are $\mathbb{R}$ and $\mathbb{R}^{\mathrm{d}}$-valued random variables, respectively. We can apply the Cauchy-Schwarz inequality to deduce that $\mathbf{E}[XY]\otimes \mathbf{E}[XY]\preceq \mathbf{E}[X^2]\mathbf{E}[Y\otimes Y]$. This is useful since it allows us to apply the bound on $\mathbf{E}[X^2]$ given in $(37)$. However, without our sampling procedure, the LHS of $(33)$ takes the form $\mathbf{E}[X^2 Y\otimes Y]$, which cannot be directly bounded because the random variable $X$ is not almost surely bounded. Also, the inequality $\mathbf{E}[X^2 Y\otimes  Y]\preceq \mathbf{E}[X^2]\mathbf{E}[Y\otimes Y]$ is not true in general. This negates the possibility of applying the bound of $\mathbf{E}[X^2]$ to upper bound the LHS of $(33)$.
>
> **References:**
>
> [1] Ilyas Fatkhullin, Anas Barakat, Anastasia Kireeva, and Niao He. Stochastic policy gradient methods: Im-
> proved sample complexity for Fisher-non-degenerate policies. International Conference on Machine
> Learning, pp. 9827–9869. PMLR, 2023.
>
> [2] Washim Uddin Mondal, Vaneet Aggarwal, and Satish Ukkusuri. Mean-field control-based approximation of multi-agent reinforcement learning in the presence of a non-decomposable shared global state. Transactions
> on Machine Learning Research, 2023.
>
> [3] Lingxiao Wang, Qi Cai, Zhuoran Yang, and Zhaoran Wang. Neural policy gradient methods: Global optimality and rates of convergence. In International Conference on Learning Representations, 2020.

---

> > ### Comment · Reviewer_4KU4 · 2026-04-13
> >
> > I thank the authors for the detailed response. I think my concerns have been properly addressed. I have no further questions.

---

### Review · Reviewer_sPN4 · 2026-03-08

**Summary Of Contributions:**

The paper introduces **PDR-ANPG** (Primal-Dual based Regularized Accelerated Natural Policy Gradient), an algorithm designed to solve Constrained Markov Decision Processes (CMDPs) using general parameterized policies. The primary contribution is achieving **last-iterate** convergence guarantees, which is a more stringent requirement than the standard average-iterate convergence, while improving the sample complexity to $\\tilde{O}(\\epsilon^{-2}\\min\\{\\epsilon^{-2}, \\epsilon_{\\rm{bias}}^{-1/3}\\})$.

### Strengths

* **Significant Complexity Improvement**: The leap from $\\epsilon^{−7}$ to $\\epsilon^{−4}$ for last-iterate convergence in the general parameterization setting is a substantial theoretical contribution.
* **Clarity of Theoretical Intuition**: The authors do an excellent job of "translating" complex propositions into accessible insights, making the theoretical machinery easier to follow.
* **Novel Sampling Scheme**: The use of an additional expectation over $a\\sim\\pi_\\theta(\\hat{s})$ in the gradient estimation is a clever technical contribution that allows the authors to bound the left-hand side of (33) from above.

### Weaknesses

* **Lack of Empirical Results**: The absence of even a simple toy simulation makes it difficult to assess if the theoretical gains translate to practical performance or if the constant factors are prohibitively large.
* **Constraint Limitations**: The theoretical framework is restricted to a single constraint. In many practical CMDP settings, agents must satisfy multiple safety or resource constraints simultaneously, but the extension to multiple constraints is not discussed.
* **Inconsistency in State Space Definitions**: There is a technical discrepancy between the main text and the appendix regarding the nature of the environment. The state space is assumed to be countable in Section 2, but the proofs in Section A treat it as a finite set.
* **Safety Guarantees in Expectation vs. Per-Run**: The algorithm's safety convergence is established only in an expected sense, rather than providing high-probability or per-run guarantees. While the authors achieve an ϵ constraint violation in expectation, most prior works introduced in the paper, such as Ying et al. (2022), Gladin et al. (2023), and Ding et al. (2024), offer robust per-run safety assurances. The lack of these stronger guarantees limits the algorithm's practical reliability, as individual executions may still experience significant violations.
* **Typesetting and Formatting Issues**: The manuscript contains several presentation errors, most notably superfluous line-breaks within equations. These artifacts, likely resulting from a template transition, occasionally hinder the readability of the mathematical arguments and give the paper an unpolished appearance.

**Audience:**

Yes

**Audience Explanation:**

The findings of this paper are of significant interest to the TMLR's audiences as they address a major open challenge in safe reinforcement learning: achieving tight convergence guarantees for general parameterized policies. The proposed algorithm PDR-ANPG improves the sample complexity from the previous state-of-the-art of $\\tilde{O}(\\epsilon^{−7})$ to $\\tilde{O}(\\epsilon^{−4})$. Furthermore, the focus on last-iterate convergence, rather than average-iterate, aligns with the needs of practitioners who require stable, high-performing policies at the end of training. The theoretical techniques introduced also offer valuable insights that are likely to influence future work in constrained RL.

**Broader Impact Concerns:**

The current submission does not include a Broader Impact Statement; however, given that the nature of the work is purely theoretical and focuses on establishing convergence guarantees for general parametrised policies for CMDPs, I do not believe a formal statement is strictly necessary. The research does not involve human subjects, real-world datasets with privacy concerns, or immediate applications that pose ethical risks. While the algorithm is designed for 'safe' reinforcement learning, its current form is a mathematical contribution to the foundations of the field rather than a deployed system with direct societal implications.

**Claims And Evidence:**

Yes

**Claims Explanation:**

The claims made in the submission are supported by a logically consistent and mathematically sound framework presented in the main text. While I have not performed a line-by-line verification of the full proofs in the appendix, the evidence and derivations provided in the main paper are convincing and provide sufficient detail to justify the claimed sample complexity improvements.

**Requested Changes:**

The following requested changes are non-critical but would significantly strengthen the submission and improve its clarity:

## Technical and Content Improvements

1. **Empirical Validation**: While the primary contribution is theoretical, adding empirical results on a simple toy environment would be highly beneficial.
2. **Consistency in Environment Definitions**: Please ensure the definition of the state space is consistent throughout the manuscript. Currently, Section 2 assumes a countable state space, while the proof in Section A assume a finite set.
3. **Clarification of Lemma 7 and Equation (37)**: The presentation of Equation (37) is somewhat misleading, as it appears to be a consequence of Lemma 7. In the revision, please clarify that this equation is part of the proof sketch for the lemma rather than a separate result derived from it.

## Typesetting and Formatting Improvements

1. **Equation Formatting**: The manuscript contains numerous superfluous line breaks within mathematical expressions, likely an artifact of a template transition. Please reformat these to improve the flow and readability of the proofs.
2. **Layout and Spacing**: On page 5, the vertical white space between the equations and the body text is excessively large. Please adjust the LaTeX spacing to ensure a more professional layout.
3. **Reference Corrections**:
    * In the sentence preceding Lemma 3 on page 5, the text incorrectly refers to "Lemma 2" when it should refer to "Lemma 3."
    * On page 11, the phrase "update equations equation 21−equation 24" should be corrected to "update equations 21–24.".
4. **Algorithm Typesetting**: The comments within the algorithms currently use a "/* ... /*" style. It is recommended to use the standard \Comment macro (or the equivalent in your specific algorithm package) to ensure the pseudocode adheres to standard academic formatting.

---

> ### Author Response · Authors · 2026-03-31
> **Response to Reviewer sPN4**
>
> **Multiple Constraints:** The CMDP setup with multiple constraints is a natural extension of our framework. However, since our primary goal is to develop a novel technique to massively improve the state-of-the-art sample-complexity bounds for CMDPs with single constraints, we believe generalizing our analysis to multiple constraints would overly complicate the proofs and may obscure our primary contribution from readers. We have enlisted it as a future work in the conclusion section.
>
> **High-probability Guarantee:** Obtaining a high-probability guarantee, instead of an expectation-based one, is an ambitious next step of our work. However, as stated in our paper, even establishing a near-optimal expectation-based last-iterated guarantee is still an open problem for general parameterized CMDPs. We have listed deriving high-probability guarantees as future work in the conclusion section.
>
> **On Countable State Space:** We would like to emphasize that Lemma 2 is a slightly modified version of the celebrated policy gradient theorem, and is not one of our primary contributions. Its proof is intended to establish the result for the simplest possible case: a finite-state space $\mathcal{S}$. The result also holds in general. For countably infinite $\mathcal{S}$, the finite sum on $\mathcal{S}$ has to be replaced by an infinite sum. Similarly, for a general $\mathcal{S}$, the sum needs to be replaced by an appropriate integral. However, in these cases, we need to specify sufficient conditions that ensure the convergence of the infinite sum or the integral. Such technicalities may over-complicate the proof and hinder the main message from the readers.
>
> **On Experiments:** Since our primary objective is to improve the state-of-the-art sample-complexity bounds, we have primarily focused on theoretical innovations. However, we agree that well-designed experiments could corroborate our theoretical findings. We will perform these experiments in the future to strengthen our claims.
>
> **Clarification on Lemma 7 and Eq. (37):** Thank you for the suggestion. In the revised manuscript, we have mentioned that Eq. (37) is a part of the proof sketch of Lemma 7.
>
> **On Typesetting and Formatting Issues:** Thank you for the suggestions. We have made all the suggested changes in the revised paper.

---

### Review · Reviewer_k7va · 2026-03-20

**Summary Of Contributions:**

This paper presents a new algorithm called PDR-ANPG to solve Constrained Markov Decision Processes (CMDPs) with general policy parameterization. The core contribution is achieving last-iterate convergence, ensuring the very last policy generated is near-optimal and safe, rather than just the average of all policies, which is critical for safety-sensitive applications like autonomous driving.

**Audience:**

Yes

**Audience Explanation:**

Gemini said TMLR's audience of reinforcement learning specialists would find these findings significant because the paper provides a major improvement in last-iterate guarantees for general parameterized policies, which are vital for safety-sensitive applications.

**Claims And Evidence:**

Yes

**Claims Explanation:**

The paper achieves a last-iterate $\epsilon$ optimality gap and $\epsilon$ constraint violation for general parameterized CMDPs with a sample complexity of $\tilde{\mathcal{O}}(\epsilon^{-2} \min\{\epsilon^{-2}, \epsilon_{bias}^{-1/3}\})$ , which improves upon the previous state-of-the-art $\tilde{\mathcal{O}}(\epsilon^{-7})$. To support this result, it proposes the PDR-ANPG algorithm using entropy and quadratic regularizers to stabilize last-iterate convergence ; introduces a novel sampling procedure that uses an additional expectation over actions to ensure advantage estimates remain bounded ; and leverages an ASGD subroutine to ensure the Natural Policy Gradient estimator bias decays exponentially rather than polynomially. This theoretical argument is convincing.

While the theoretical framework appears sound, incorporating numerical experiments to validate the proposed algorithm's superiority over baseline methods would be highly beneficial . Empirical results would significantly strengthen the paper's overall solidity and practical impact.

**Requested Changes:**

1. Gap from Lower Bounds: While this work significantly improves the sample complexity to $\tilde{\mathcal{O}}(\epsilon^{-4})$ for complete policies, a notable gap remains when compared to the theoretical lower bound of $\Omega(\epsilon^{-2})$. Closing this gap represents a critical frontier for future research in general parameterized CMDPs. Is the $\Omega(\epsilon^{-2})$ bound attainable in this setting, and what are the primary technical obstacles to achieving it?

2. Knowledge of Approximation Error: The choice of critical hyperparameters, such as the learning rate and the number of iterations, depends on the value of $\epsilon_{bias}$. In practice, this transferred compatibility error is often unknown and difficult to estimate accurately for complex policy classes. The theory would be strengthened by developing a version of the algorithm that does not require prior knowledge of $\epsilon_{bias}$ to set optimal hyperparameters.

3. To strengthen the submission, it is essential to provide empirical evidence for validating the theoretical breakthroughs claimed in the paper.

---

> ### Author Response · Authors · 2026-03-31
> **Response to Reviewer k7va**
>
> **Barrier to Lower Bound:** Our sample complexity bound to ensure $\mathcal{O}(\epsilon+(\epsilon\_{\mathrm{bias}})\^{\frac{1}{6}})$ optimality gap and constraint violation is $\tilde{\mathcal{O}}(\epsilon\^{-2}\min\\{\epsilon\^{-2}, \epsilon\_{\mathrm{bias}}\^{-\frac{1}{3}}\\})$. Therefore, if $\epsilon\_{\mathrm{bias}}>0$ is independent of $\epsilon$, then we already obtain a sample complexity of $\tilde{\mathcal{O}}(\epsilon\^{-2})$. However, if $\epsilon_{\mathrm{bias}}=0$, then the sample complexity turns out to be $\tilde{\mathcal{O}}(\epsilon\^{-4})$. Our current techniques, to the best of our understanding, seem incapable of improving the sample complexity further. It appears from Eq. (94), (95), and (103) that either a very large or very small value of the regularization parameter $\tau$ has a detrimental effect on the optimality gap and constraint violation. In the sweet spot away from the extremities, the optimal choice of $\tau$ leads to our current results. Breakthrough ideas in analysis and/or algorithmic techniques are needed to move the current state of the art closer to the theoretical lower bound.
>
> **On knowledge of $\epsilon\_{\mathrm{bias}}$:** As mentioned in our paper, some of the hyperparameters of our algorithm indeed depend on $\epsilon\_{\mathrm{bias}}$. We also agree that finding the exact value of $\epsilon\_{\mathrm{bias}}$ might be difficult in practice,  although it might be possible to obtain an upper bound, $\bar{\epsilon}\_{\mathrm{bias}}$, of the desired quantity. For example, [1] shows that for a two-layer neural parameterization of width $m$, the function approximation error is $\mathcal{O}(m\^{-1/8})$ (see Theorem A.4 and the subsequent discussion). Our algorithm still works if this bound is used as a proxy of $\epsilon\_{\mathrm{bias}}$. In this case, the convergence error and the sampling complexity will be functions of $\bar{\epsilon}\_{\mathrm{bias}}$, instead of $\epsilon\_{\mathrm{bias}}$. In the absence of appropriate upper bounds, we can use hyperparameter tuning, a common method for setting unknown hyperparameters for machine learning algorithms. We agree with the reviewer's vision of designing parameter-free algorithms. However, such a task seems beyond the reach of current theoretical techniques, at least for obtaining last-iterate guarantees for general parameterized CMDPs.
>
> **On Experiments:** Since our primary objective is to improve the state-of-the-art sample-complexity bounds, we have focused primarily on theoretical innovations. However, we agree that well-designed experiments could corroborate our theoretical findings. We will perform these experiments in the future to strengthen our claims.
>
> **References:**
>
> [1] Lingxiao Wang, Qi Cai, Zhuoran Yang, and Zhaoran Wang. Neural policy gradient methods: Global optimality and rates of convergence. In International Conference on Learning Representations, 2020.

---

### Review · Reviewer_MZ4o · 2026-03-23

**Summary Of Contributions:**

# Summary Of Contributions
The paper asks the question of whether an algorithm can be designed for finding optimal 'general parameterised' policies with last-iterate guarantees in Constrained Markov Decision Processes that beat the state-of-the-art of sample complexity of $O(e^{-7})$ and answers it with a yes, having achieved $O(e^{-4})$. These are achieved by proposing an algorithm (algorithm 2) that uses entropy and quadratic regularisers as well as a sampling procedure (algorithm 1) that can efficiently sample.

The paper provides a theoretical analysis of the proposed algorithms to show that they achieve the claimed sample efficiency.


# Strengths And Weaknesses

### Strengths
- The writing is generally great, with clear exposition on the interpretation of the math and algorithms. I was particularly grateful for the second paragraph on page 6.
- The flow of the mathematical argument is organised well, making it easier to parse.
- I did not rigorously analyse the proofs in the Appendix. However, the reasoning in the main text seems sufficiently sound.

#### Weaknesses
- I believe that the conclusion could be expanded to include a discussion / summary of practical applications of where the results of the paper are applicable and not applicable.
- I noted that there is not experimental results reported in the paper. Given that the paper presents novel algorithms and theoretical analysis of it, some *simple* experiments showcasing the expected trends would be welcome.
- When assumptions are mentioned, it would make sense to write a sentence to illustrate why the assumption is useful / necessary and / or write examples of pathological cases and why they do not occur commonly(?). See the Requested Changes section for more specific questions.

**Audience:**

Yes

**Audience Explanation:**

The paper presents a significant improvement over existing results, along with a clear and organised argument and analysis, which I believe will be of interest to TMLR's audience.

**Claims And Evidence:**

Yes

**Claims Explanation:**

The paper is written well with a clear organisation and discussion of the core mathematical analysis and argument. To the best of my knowledge, the mathematical argument is sound in the main text, although I did not review the proofs in the Appendix rigorously.

**Requested Changes:**

Please note that most of these are minor changes to consistency.
### Section 2
- Why is '(stationary)' in brackets? Do you mean 'possibly stationary'?
- Second paragraph, there is a repeated 'the' in '... the the advantage ...'.
- Could the authors please write a short sentence in the first paragraph explaining when a utility function is used as opposed to a reward or cost function? Is the utility function simply written as a generalisation of the latter two?
- 'The expected value of $V_g^\pi(s)$ obtained over $s \sim \rho$ is denoted as $J_g^\pi$' — why not write this as an equation as well, i.e., $J_g^\pi = \textbf{E}_{s \sim \rho}\left[V_g^\pi(s)\right]$ .

### Section 3
- Are the curly brackets in Equation 7 denoting something other than grouping? If so, I think regular brackets would suffice here. If not, could, the notation be explained since this seems non-standard.
- Paragraph before Lemma 3, 'Fortunately, Lemma 2...' — I think the authors meant 'Lemma 3'.
- '... carefully chosen set of ...' — could a description / footnote of how these are chosen be written. Furthermore, where '$\lambda_\mathrm{max}$ is stated later' is written, please add a concrete pointer (i.e., which section, equation, etc) to it.
- Whenever 'as explained later', or 'is stated later', or otherwise is used, please specifically state where so that it is easy to find (see example above as well as in the first paragraph on page 7).
- Please use `\mathrm` when writing text in equations (e.g., equation 15 and equation 19). Alternatively, the authors could write a regular paragraph for 'where', and the follow that with another equation.

### Section 4
- Assumption 2 — are there examples of when this assumption does not hold? Are these pathological examples rare?
- Assumption 3 — what happens to the analysis when this assumption isn't true?
- Paragraph after Assumption 4 '... implies that said function ...' — which function?
- Paragraph after Assumption 4 'Later, we shall see ...' — Please be more specific about where.
- Paragraph after Assumption 4 — Could the authors please add a sentence stating why it makes sense for this assumption to be commonplace?

---

> ### Author Response · Authors · 2026-03-31
> **Response to Reviewer MZ4o**
>
> **Minor Comments:** Thank you for the suggestions and pointing out the typos. We have incorporated all of the suggested changes in the revised paper. The modifications have been highlighted in blue.
>
> **Applicability of the Results:** Our results apply to those cases where the action space is finite. Extending it to infinite action spaces would be an interesting future work. We have added a brief discussion on it in the conclusion section.
>
> **On Stationary Policies:** The term stationary is written in brackets to emphasize that we have chosen the policies to be stationary, although in general, a policy could be non-stationary. The reason behind this choice is the well-known result that an optimal policy for an MDP must be stationary [3].
>
> **On Utility Function:** The utility function is defined to be a general function of the form $g: \mathcal{S}\times \mathcal{A}\rightarrow \mathbb{R}$. Since both reward and cost functions are of this form, they can be thought of as utility functions. Moreover, in Algorithm 1, we estimate the advantage function corresponding to $g=r+\lambda c+ \tau \psi\_{\theta}$. Here, the function $g$ also falls under the umbrella of utility functions. Therefore, although the utility function generalizes the concepts of reward and cost functions, its application is not restricted solely to them.
>
> **On Assumption 2:** Assumption 2 essentially dictates that neither the policy function nor its gradient should rapidly change as a result of a slight nudge in the $\theta$ parameter. In the absence of such a restriction, it will be difficult for the policy gradient algorithm to rapidly converge to a stationary point. We have added this discussion in the revised manuscript.
>
> **On Assumption 3:** Assumption 3 introduces the term $\epsilon\_{\mathrm{bias}}$, which can be thought of as the expressivity power of the parameterized policy class. For rich policy classes (such as neural network-based policies with sufficiently large width), $\epsilon\_{\mathrm{bias}}$ is insignificantly small. The effect of $\epsilon\_{\mathrm{bias}}$ can be seen in the optimality gap and constraint violation, which are $\mathcal{O}(\epsilon + (\epsilon\_{\mathrm{bias}})\^{\frac{1}{6}})$, as stated in Theorem 1. Therefore, for inappropriate choice of policy classes, the value of $\epsilon\_{\mathrm{bias}}$ will be large, leading to poor optimality gap and constraint violation guarantees.
>
> **On Assumption 4:** Assumption 4 concerns the Fisher matrix $F(\theta) = \mathbf{E}\_{(s, a)\sim \nu^{\pi_\theta}}[\nabla\_{\theta}\log \pi\_{\theta}(a|s)\otimes \nabla\_{\theta}\log \pi\_{\theta}(a|s)]$. The very definition of $F(\theta)$ ensures that $F(\theta)$ is positive semi-definite, i.e., all its eigenvalues are non-negative. Assumption 4 adds the mild restriction that, not only the eigenvalues of $F(\theta)$ have to be non-negative, they must be strictly larger than a positive quantity $\mu$. This assumption, popularly known as the Fisher non-degeneracy, is central to policy gradient algorithms, as evident by its widespread usage in the literature. Many policy classes, such as Cauchy, Gaussian, and exponential with parameterized mean, etc., satisfy this assumption [1]. It is worth noting that softmax policies may not satisfy this assumption if they are close to being deterministic. However, if the softmax parameters are bounded in a finite interval, then they can be shown to satisfy this assumption (Remark 1 in [2]). A detailed discussion on this topic has been added in the revised manuscript.
>
> **On Experiments:** Since our primary objective is to improve the state-of-the-art sample complexity bounds, we have mainly focused on theoretical innovations. However, we agree that well-designed experiments could corroborate our findings. We will perform these experiments in the future to strengthen our claims.
>
> **References:**
>
> [1] Ilyas Fatkhullin, Anas Barakat, Anastasia Kireeva, and Niao He. "Stochastic policy gradient methods: Im-
> proved sample complexity for Fisher-non-degenerate policies". International Conference on Machine
> Learning, pp. 9827–9869. PMLR, 2023.
>
> [2] Washim Uddin Mondal, Vaneet Aggarwal, and Satish Ukkusuri. Mean-field control based approximation of multi-agent reinforcement learning in presence of a non-decomposable shared global state. Transactions
> on Machine Learning Research, 2023.
>
> [3] Puterman, Martin L. "Markov decision processes." Handbooks in operations research and management science 2 (1990): 331-434.

---

### Decision · Action_Editor_MqZQ · 2026-04-27

**Recommendation:** Accept as is

**Audience:**

Yes

**Audience Explanation:**

The paper studies constrained Markov decision processes (CMDPs) under general parameterized policies and establishes convergence guarantees for primal-dual natural policy gradient algorithms. This safe reinforcement learning setting is a central problem in the machine learning community, with broad applications in areas such as robotics, autonomous driving, and other safety-critical decision-making systems. Developing rigorous convergence theory for this problem is important not only for advancing theoretical understanding, but also for providing guidance on which algorithmic designs are most reliable and effective in practice. The theoretical contributions of this work will serve a great value to the community.

**Claims And Evidence:**

Yes

**Claims Explanation:**

The paper has provided the proofs for the theoretical results claimed in the paper.